# Augmented Sliced Wasserstein Distances

## Abstract

While theoretically appealing, the application of the Wasserstein distance to large-scale machine learning problems has been hampered by its prohibitive computational cost. The sliced Wasserstein distance and its variants improve the computational efficiency through the random projection, yet they suffer from low accuracy if the number of projections is not sufficiently large, because the majority of projections result in trivially small values. In this work, we propose a new family of distance metrics, called augmented sliced Wasserstein distances (ASWDs), constructed by first mapping samples to higher-dimensional hypersurfaces parameterized by neural networks. It is derived from a key observation that (random) linear projections of samples residing on these hypersurfaces would translate to much more flexible *nonlinear* projections in the original sample space, so they can capture complex structures of the data distribution. We show that the hypersurfaces can be optimized by gradient ascent efficiently. We provide the condition under which the ASWD is a valid metric and show that this can be obtained by an injective neural network architecture. Numerical results demonstrate that the ASWD significantly outperforms other Wasserstein variants for both synthetic and real-world problems.

## 1   Introduction

Comparing samples from two probability distributions is a fundamental problem in statistics and machine learning. The optimal transport (OT) theory [Villani, 2008] provides a powerful and flexible theoretical tool to compare degenerative distributions by accounting for the metric in the underlying spaces. The Wasserstein distance, which arises from the optimal transport theory, has become an increasingly popular choice in various machine learning domains ranging from generative models to transfer learning [Gulrajani et al., 2017; Arjovsky et al., 2017; Kolouri et al., 2019b; Cuturi and Doucet, 2014; Courty et al., 2016].

Despite its favorable properties, such as robustness to disjoint supports and numerical stability [Arjovsky et al., 2017], the Wasserstein distance suffers from high computational complexity especially when the sample size is large. Besides, the Wasserstein distance itself is the result of an optimization problem — it is non-trivial to be integrated into an end-to-end training pipeline of deep neural networks, unless one can make the solver for the optimization problem differentiable. Recent advances in computational optimal transport methods focus on alternative OT-based metrics that are computationally efficient and solvable via a differentiable optimizer [Peyré and Cuturi, 2019]. Entropy regularization is introduced in the Sinkhorn distance [Cuturi, 2013] and its variants [Altschuler et al., 2017; Dessein et al., 2018] to smooth the optimal transport problem; as a result, iterative matrix scaling algorithms can be applied to provide significantly faster solutions with improved sample complexity [Genevay et al., 2019].

An alternative approach is to approximate the Wasserstein distance through *slicing*, i.e. linearly projecting, the distributions to be compared. The sliced Wasserstein distance (SWD) [Bonneel et al., 2015] is defined as the expected value of Wasserstein distances between one-dimensional random projections of high-dimensional distributions. The SWD shares similar theoretical properties with the

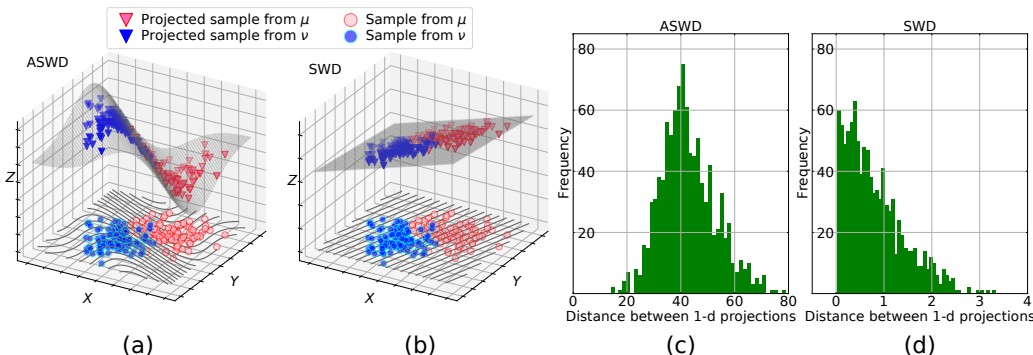

Figure 1: (a) and (b) are visualizations of projections for the ASWD and the SWD between two 2-dimensional Gaussians. (c) and (d) are distance histograms for the ASWD and the SWD between two 100-dimensional Gaussians. Figure 1(a) shows that the injective neural network embedded in the ASWD learns data patterns (in the $X$-$Y$ plane) and produces well-separate projected values ($Z$-axis) between distributions in a random projection direction. The high projection efficiency of the ASWD is evident in Figure 1(c), as almost all random projection directions in a 100-dimensional space lead to significant distances between 1-dimensional projections. In contrast, random linear mappings in the SWD often produce closer 1-d projections ($Z$-axis) (Figure 1(b)); as a result, a large percentage of random projection directions in the 100-d space result in trivially small distances (Figure 1(d)), leading to a low projection efficiency in high-dimensional spaces.

Wasserstein distance [Bonnotte, 2013] and is computationally efficient since the Wasserstein distance in one-dimensional space has a closed-form solution based on sorting. [Deshpande et al., 2019] extends the sliced Wasserstein distance to the max-sliced Wasserstein distance (Max-SWD), by finding a single projection direction with the maximal distance between projected samples. The subspace robust Wasserstein distance extends the idea of slicing to projecting distributions on linear subspaces [Paty and Cuturi, 2019]. However, the linear nature of these projections usually leads to low projection efficiency of the resulted metrics in high-dimensional spaces [Deshpande et al., 2019; Kolouri et al., 2019a].

Different variants of the SWD have been proposed to improve the projection efficiency of the SWD, either by introducing nonlinear projections or by optimizing the distribution of random projections. Specifically, [Kolouri et al., 2019a] extends the connection between the sliced Wasserstein distance and the Radon transform [Radon, 1917] to introduce generalized sliced Wasserstein distances (GSWDs) by utilizing generalized Radon transforms (GRTs), which are defined by nonlinear defining functions and lead to nonlinear projections. A variant named the GSWD-NN was proposed in [Kolouri et al., 2019a] to generate nonlinear projections *directly* with neural network outputs, but it does not fit into the theoretical framework of the GSWD and does not guarantee a valid metric. In contrast, the distributional sliced Wasserstein distance (DSWD) and its nonlinear version, the distributional generalized sliced Wasserstein distance (DGSWD), improve their projection efficiency by finding a distribution of projections that maximizes the expected distances over these projections. The GSWD and the DGSWD exhibit higher projection efficiency than the SWD in the experiment evaluation, yet they require the specification of the particular form of defining functions from a limited class of candidates. However, the selection of defining functions is usually a task-dependent problem and requires domain knowledge, and the impact on performance from different defining functions is still unclear.

In this paper, we present the augmented sliced Wasserstein distance (ASWD), a distance metric constructed by first mapping samples to hypersurfaces in an *augmented* space, which enables flexible nonlinear slicing of data distributions for improved projection efficiency (See Figure 1). Our main contributions include: (i) We exploit the capacity of nonlinear projections employed in the ASWD by constructing injective mapping with arbitrary neural networks; (ii) We prove that the ASWD is a valid distance metric; (iii) We provide a mechanism in which the hypersurface where high-dimensional distributions are projected onto can be optimized and show that the optimization of hypersurfaces can help improve the projection efficiency of slice-based Wasserstein distances. Hence, the ASWD is data-adaptive, i.e. the hypersurfaces can be learned from data. This implies one does not need to manually design a function from the limited class of candidates; (iv) We demonstrate superior performance of the ASWD in numerical experiments for both synthetic and real-world datasets.

The remainder of the paper is organized as follows. Section 2 reviews the necessary background. We present the proposed method and its numerical implementation in Section 3. Related work are discussed in Section 4. Numerical experiment results are presented and discussed in Section 5. We conclude the paper in Section 6.

## 2 Background

In this section, we provide a brief review of concepts related to the proposed work, including the Wasserstein distance, (generalized) Radon transform and (generalized) sliced Wasserstein distances.

**Wasserstein distance:** Let $P_k(\Omega)$ be a set of Borel probability measures with finite $k$-th moment on a Polish metric space $(\Omega, d)$ [Villani, 2008]. Given two probability measures $\mu, \nu \in P_k(\Omega)$, whose probability density functions (PDFs) are $p_\mu$ and $p_\nu$, the Wasserstein distance of order $k \in [1, +\infty)$ between $\mu$ and $\nu$ is defined as:

$$W_k(\mu, \nu) = \left( \inf_{\gamma \in \Gamma(\mu, \nu)} \int_\Omega d(x, y)^k d\gamma(x, y) \right)^{\frac{1}{k}}, \tag{1}$$

where $d(\cdot, \cdot)^k$ is the cost function, $\Gamma(\mu, \nu)$ represents the set of all transportation plans $\gamma$, i.e. joint distributions whose marginals are $p_\mu$ and $p_\nu$, respectively. With a slight abuse of notation, we interchangeably use $W_k(\mu, \nu)$ and $W_k(p_\mu, p_\nu)$.

While the Wasserstein distance is generally intractable for high-dimensional distributions, there are several favorable cases where the optimal transport problem can be efficiently solved. If $\mu$ and $\nu$ are continuous one-dimensional measures defined on a linear space equipped with the $L^k$ norm, the Wasserstein distance between $\mu$ and $\nu$ has a closed-form solution [Peyré and Cuturi, 2019]:

$$W_k(\mu, \nu) = \left( \int_0^1 |F_\mu^{-1}(z) - F_\nu^{-1}(z)|^k dz \right)^{\frac{1}{k}}, \tag{2}$$

where $F_\mu^{-1}$ and $F_\nu^{-1}$ are inverse cumulative distribution functions (CDFs) of $\mu$ and $\nu$, respectively.

**Radon transform and generalized Radon transform:** The Radon transform [Radon, 1917] maps a function $f(\cdot) \in L^1(\mathbb{R}^d)$ to the space of functions defined over spaces of lines in $\mathbb{R}^d$. The Radon transform of $f(\cdot)$ is defined by line integrals of $f(\cdot)$ along all possible hyperplanes in $\mathbb{R}^d$:

$$\mathcal{R}f(t, \theta) = \int_{\mathbb{R}^d} f(x)\delta(t - \langle x, \theta \rangle)dx, \tag{3}$$

where $t \in \mathbb{R}$ and $\theta \in \mathbb{S}^{d-1}$ represent the parameters of hyperplanes $\{x \in \mathbb{R}^d \mid \langle x, \theta \rangle = t\}$, $\delta(\cdot)$ is the Dirac delta function, and $\langle \cdot, \cdot \rangle$ refers to the Euclidean inner product.

By replacing the inner product $\langle x, \theta \rangle$ in Equation (3) with $\beta(x, \theta)$, a specific family of functions named as *defining function* in [Kolouri et al., 2019a], the generalized Radon transform (GRT) [Beylkin, 1984] is defined as integrals of $f(\cdot)$ along hypersurfaces defined by $\{x \in \mathbb{R}^d \mid \beta(x, \theta) = t\}$:

$$\mathcal{G}f(t, \theta) = \int_{\mathbb{R}^d} f(x)\delta(t - \beta(x, \theta))dx, \tag{4}$$

where $t \in \mathbb{R}$, $\theta \in \Omega_\theta$ while $\Omega_\theta$ is a compact set of all feasible $\theta$, e.g. $\Omega_\theta = \mathbb{S}^{d-1}$ for $\beta(x, \theta) = \langle x, \theta \rangle$ [Kolouri et al., 2019a].

In practice, we can empirically approximate the Radon transform and the GRT of a probability density function $p_\mu$ via:

$$\mathcal{R}p_\mu(t, \theta) \approx \frac{1}{N} \sum_{n=1}^{N} \delta(t - \langle x_n, \theta \rangle), \tag{5}$$

$$\mathcal{G}p_\mu(t, \theta) \approx \frac{1}{N} \sum_{n=1}^{N} \delta(t - \beta(x_n, \theta)), \tag{6}$$

where $x_n \sim p_\mu$ and $N$ is the number of samples. Notably, the Radon transform is a linear bijection [Helgason, 1980], and the GRT is a bijection if the defining function $\beta$ satisfies certain conditions [Beylkin, 1984].

106 **Sliced Wasserstein distance and generalized sliced Wasserstein distance:** By applying the Radon
107 transform to $p_\mu$ and $p_\nu$ to obtain multiple projections, the sliced Wasserstein distance (SWD) decom-
108 poses the high-dimensional Wasserstein distance into multiple one-dimensional Wasserstein distances
109 which can be efficiently evaluated [Bonneel et al., 2015]. The $k$-SWD between $\mu$ and $\nu$ is defined by:

$$\text{SWD}_k(\mu,\nu) = \left( \int_{\mathbb{S}^{d-1}} W_k^k \big( \mathcal{R}p_\mu(\cdot,\theta), \mathcal{R}p_\nu(\cdot,\theta) \big) d\theta \right)^{\frac{1}{k}}, \tag{7}$$

110 where the Radon transform $\mathcal{R}$ defined by Equation (3) is adopted as the measure push-forward operator.
111 The GSWD generalizes the idea of SWD by slicing distributions with hypersurfaces rather than
112 hyperplanes [Kolouri et al., 2019a]. The GSWD is defined as:

$$\text{GSWD}_k(\mu,\nu) = \left( \int_{\Omega_\theta} W_k^k \big( \mathcal{G}p_\mu(\cdot,\theta), \mathcal{G}p_\nu(\cdot,\theta) \big) d\theta \right)^{\frac{1}{k}}, \tag{8}$$

113 where the GRT $\mathcal{G}$ is used as the measure push-forward operator. The Wasserstein distances between
114 one-dimensional distributions can be obtained by sorting projected samples and calculating the
115 distance between sorted samples [Kolouri et al., 2019b]: with $L$ random projections, the SWD and
116 GSWD between $\mu$ and $\nu$ can be approximated by:

$$\text{SWD}_k(\mu,\nu) \approx \left( \frac{1}{NL} \sum_{l=1}^{L} \sum_{n=1}^{N} |\langle x_{I_x^l[n]}, \theta_l \rangle - \langle y_{I_y^l[n]}, \theta_l \rangle|^k \right)^{\frac{1}{k}}, \tag{9}$$

$$\text{GSWD}_k(\mu,\nu) \approx \left( \frac{1}{NL} \sum_{l=1}^{L} \sum_{n=1}^{N} |\beta(x_{I_x^l[n]}, \theta_l) - \beta(y_{I_y^l[n]}, \theta_l)|^k \right)^{\frac{1}{k}}, \tag{10}$$

117 where $I_x^l$ and $I_y^l$ are sequences consisting of the indices of sorted samples which satisfy $\langle x_{I_x^l[n]}, \theta_l \rangle \leq$
118 $\langle x_{I_x^l[n+1]}, \theta_l \rangle$, $\langle y_{I_y^l[n]}, \theta_l \rangle \leq \langle y_{I_y^l[n+1]}, \theta_l \rangle$ in the SWD, and $\beta(x_{I_x^l[n]}, \theta_l) \leq \beta(x_{I_x^l[n+1]}, \theta_l)$,
119 $\beta(y_{I_y^l[n]}, \theta_l) \leq \beta(y_{I_y^l[n+1]}, \theta_l)$ in the GSWD. It is proved in [Bonnotte, 2013] that the SWD is a valid
120 distance metric. The GSWD is a valid metric except for its neural network variant [Kolouri et al., 2019a].

# 3 Augmented sliced Wasserstein distances

122 In this section, we propose a new distance metric called the augmented sliced Wasserstein distance
123 (ASWD), which embeds flexible nonlinear projections in its construction. We also provide an
124 implementation recipe for the ASWD.

## 3.1 Spatial Radon transform and augmented sliced Wasserstein distance

126 In the definitions of the SWD and GSWD, the Radon transform [Radon, 1917] and the generalized
127 Radon transform (GRT) [Beylkin, 1984] are used as the push-forward operator for projecting
128 distributions to a one-dimensional space. However, it is not straightforward to design defining functions
129 $\beta(x,\theta)$ [Kolouri et al., 2019a] for the GRT due to certain non-trivial requirements for the function
130 [Beylkin, 1984]. In practice, the assumption of the transform can be relaxed, as Theorem 1 shows
131 that as long as the transform is injective, the corresponding ASWD metric is a valid distance metric.

132 To help us define the augmented sliced Wasserstein distance, we first introduce the *spatial Radon*
133 *transform* which includes the vanilla Radon transform and the polynomial GRT as special cases (See
134 Remark 2).

**Definition 1.** *Given an injective mapping $g(\cdot) : \mathbb{R}^d \to \mathbb{R}^{d_\theta}$ and a probability measure $\mu \in P(\mathbb{R}^d)$*
136 *whose probability density function (PDF) is $p_\mu$, the spatial Radon transform of $p_\mu$ is defined as*

$$\mathcal{H}p_\mu(t,\theta;g) = \int_{\mathbb{R}^d} p_\mu(x)\delta(t-\langle g(x),\theta \rangle)dx, \tag{11}$$

137 *where $t \in \mathbb{R}$ and $\theta \in \mathbb{S}^{d_\theta-1}$ are the parameters of hypersurfaces $\{x \in \mathbb{R}^d \mid \langle g(x),\theta \rangle = t\}$.*

**Remark 1.** *Note that the spatial Radon transform can be interpreted as applying the vanilla Radon*
139 *transform to the PDF of $\hat{x} = g(x)$, where $x \sim p_\mu$. Denote the PDF of $\hat{x}$ by $p_{\hat{\mu}_g}$, the spatial Radon*

*transform defined by Equation (11) can be rewritten as:*

$$\begin{aligned}
\mathcal{H}p_\mu(t,\theta;g) &= E_{x \sim p_\mu}[\delta(t-\langle g(x),\theta \rangle)], \\
&= E_{\hat{x} \sim p_{\hat{\mu}_g}}[\delta(t-\langle \hat{x},\theta \rangle)] \\
&= \int p_{\hat{\mu}_g}(\hat{x})\delta(t-\langle \hat{x},\theta \rangle)d\hat{x} \\
&= \mathcal{R}p_{\hat{\mu}_g}(t,\theta).
\end{aligned} \tag{12}$$

*Hence the spatial Radon transform inherits the theoretical properties of the Radon transform subject to certain conditions of $g(\cdot)$ and incorporates nonlinear projections through $g(\cdot)$.*

In what follows, we use $f_1 \equiv f_2$ to denote functions $f_1(\cdot): X \to \mathbb{R}$ and $f_2(\cdot): X \to \mathbb{R}$ that satisfy $f_1(x) = f_2(x)$ for $\forall x \in X$.

**Lemma 1.** *Given an injective mapping $g(\cdot): \mathbb{R}^d \to \mathbb{R}^{d_\theta}$ and two probability measures $\mu,\nu \in P(\mathbb{R}^d)$ whose probability density functions are $p_\mu$ and $p_\nu$, respectively, for all $t \in \mathbb{R}$ and $\theta \in \mathbb{S}^{d_\theta-1}$, $\mathcal{H}p_\mu(t,\theta;g) \equiv \mathcal{H}p_\nu(t,\theta;g)$ if and only if $p_\mu \equiv p_\nu$, i.e. the spatial Radon transform is injective. Moreover, the spatial Radon transform is injective if and only if the mapping $g(\cdot)$ is an injection.*

See Appendix A for the proof of Lemma 1.

**Remark 2.** *The spatial Radon transform degenerates to the vanilla Radon transform when the mapping $g(\cdot)$ is an identity mapping. When $g(\cdot)$ is a homogeneous polynomial function with odd degrees, the spatial Radon transform is equivalent to the polynomial GRT [Ehrenpreis, 2003].*

Appendix B provides the proof of Remark 2.

We now introduce the augmented sliced Wasserstein distance, by utilizing the spatial Radon transform as the measure push-forward operator:

**Definition 2.** *Given two probability measures $\mu,\nu \in P_k(\mathbb{R}^d)$, whose probability density functions are $p_\mu$ and $p_\nu$, respectively, and an injective mapping $g(\cdot): \mathbb{R}^d \to \mathbb{R}^{d_\theta}$, the augmented sliced Wasserstein distance (ASWD) of order $k \in [1,+\infty)$ is defined as:*

$$\text{ASWD}_k(\mu,\nu;g) = \left( \int_{\mathbb{S}^{d_\theta-1}} W_k^k\big(\mathcal{H}p_\mu(\cdot,\theta;g),\mathcal{H}p_\nu(\cdot,\theta;g)\big)d\theta \right)^{\frac{1}{k}}, \tag{13}$$

*where $\theta \in \mathbb{S}^{d_\theta-1}$, $W_k$ is the $k$-Wasserstein distance defined by Equation (1), and $\mathcal{H}$ refers to the spatial Radon transform defined by Equation (11).*

**Remark 3.** *Following the connection between the spatial Radon transform and the vanilla Radon transform as shown in Equation (12), the ASWD can be rewritten as:*

$$\begin{aligned}
\text{ASWD}_k(\mu,\nu;g) &= \left( \int_{\mathbb{S}^{d_\theta-1}} W_k^k\big(\mathcal{R}p_{\hat{\mu}_g}(\cdot,\theta),\mathcal{R}p_{\hat{\nu}_g}(\cdot,\theta)\big)d\theta \right)^{\frac{1}{k}} \\
&= \text{SWD}_k(\hat{\mu}_g,\hat{\nu}_g),
\end{aligned} \tag{14}$$

*where $\hat{\mu}_g$ and $\hat{\nu}_g$ are probability measures on $\mathbb{R}^{d_\theta}$ which satisfy $g(x) \sim \hat{\mu}_g$ for $x \sim \mu$ and $g(y) \sim \hat{\nu}_g$ for $y \sim \nu$.*

**Theorem 1.** *The augmented sliced Wasserstein distance (ASWD) of order $k \in [1,+\infty)$ defined by Equation (13) with a mapping $g(\cdot): \mathbb{R}^d \to \mathbb{R}^{d_\theta}$ is a metric on $P_k(\mathbb{R}^d)$ if and only if $g(\cdot)$ is injective.*

The proof of Theorem 1 is provided in Appendix C. Theorem 1 shows that the ASWD is a metric given a fixed injective mapping $g(\cdot)$. In practical applications, the mapping $g(\cdot)$ needs to be optimized. We show in Corollary 1.1 that the ASWD between $\mu$ and $\nu$ with the optimized $g(\cdot)$ is also a metric under mild conditions.

**Corollary 1.1.** *The augmented sliced Wasserstein distance (ASWD) of order $k \in [1,+\infty)$ between two probability $\mu, \nu \in P_k(\mathbb{R}^d)$ defined by Equation (13) with the optimal mapping $g^*(\cdot) = \underset{g}{\arg\max}(\text{ASWD}_k(\mu,\nu;g))$ is a metric on $P_k(\mathbb{R}^d)$ when the optimization is confined to the set of bounded and injective functions $\{g(x): \mathbb{R}^d \to \mathbb{R}^{d_\theta} | \exists M \in \mathbb{R}, \forall x \in \mathbb{R}^d, ||g(x)||_2 \leq M\}$.*

The proof of Corollary 1.1 is provided in Appendix D.

**Remark 4.** *Corollary 1.1 shows that given measures $\mu_1,\mu_2,\mu_3 \in P_k(\mathbb{R}^d)$, the triangle inequality holds for the ASWD when $g(\cdot)$ is optimized for each pair of measures, as shown in Appendix D.*

 ## 3.2 Numerical implementation

179 We discuss in this section how to realize injective mapping $g(\cdot)$ with *neural networks* due to their
180 expressiveness and optimize it with gradient based methods.

181 **Injective neural networks:** As stated in Lemma 1 and Theorem 1, the injectivity of $g(\cdot)$ is the *sufficient*
182 *and necessary* condition for the ASWD being a valid metric. Thus we need specific architecture designs
183 on implementing $g(\cdot)$ by neural networks. One option is the family of invertible neural networks
184 [Behrmann et al., 2019; Karami et al., 2019], which are both injective and surjective. However, the
185 running cost of those models is usually much higher than that of vanilla neural networks. We propose
186 an alternative approach by concatenating the input $x$ of an arbitrary neural network to its output $\phi_\omega(x)$:

$$g_\omega(x) = [x, \phi_\omega(x)]. \tag{15}$$

187 It is trivial to show that $g_\omega(x)$ is injective, since different inputs will lead to different outputs. Although
188 embarrassingly simple, this idea of concatenating the input and output of neural networks has found
189 success in preserving information with dense blocks in the DenseNet [Huang et al., 2017], where the
190 input of each layer is injective to the output of all preceding layers.

191 **Optimization objective:** We aim to slice distributions with maximally discriminating hypersurfaces
192 between two distributions, so that the projected samples between distributions are most dissimilar
193 subject to certain constraints on the hypersurface, as shown in Figure 1. Similar ideas have been
194 employed to identify important projection directions [Deshpande et al., 2019; Kolouri et al., 2019a;
195 Paty and Cuturi, 2019] or a discriminative ground metric [Salimans et al., 2018] in optimal transport
196 metrics. For the ASWD, the parameterized injective neural network $g_\omega(\cdot)$ is optimized by maximizing
197 the following objective:

$$\mathcal{L}(\mu, \nu; g_\omega, \lambda) = \left( \int_{\mathbb{S}^{d_\theta - 1}} W_k^k \big( \mathcal{H}p_\mu(\cdot, \theta; g_\omega), \mathcal{H}p_\nu(\cdot, \theta; g_\omega) \big) d\theta \right)^{\frac{1}{k}} - L_\lambda, \tag{16}$$

198 where $\lambda > 0$ and the regularization term $L_\lambda = \lambda \mathbb{E}_{x,y \sim \mu, \nu} \big[ (||g_\omega(x)||_2 + ||g_\omega(y)||_2) \big]$ is used to control
199 the norm of the output of $g_\omega(\cdot)$, otherwise the projections may be arbitrarily large.

200 **Remark 5.** *The regularization coefficient $\lambda$ adjusts the introduced non-linearity in the evaluation of*
201 *the ASWD by controlling the norm of $\phi_\omega(\cdot)$ in Equation (15). In particular, when $\lambda \to \infty$, the nonlinear*
202 *term $\phi_\omega(\cdot)$ shrinks to 0. The rank of the augmented space is hence explicitly controlled by the flexible*
203 *choice of $\phi_\omega(\cdot)$ and implicitly regularized by $L_\lambda$.*

204 By plugging the optimized $g_{\omega,\lambda}^*(\cdot) = \underset{g_\omega}{\mathrm{argmax}}(\mathcal{L}(\mu, \nu; g_\omega, \lambda))$ into Equation (13), we obtain the
205 empirical version of the ASWD. Pseudocode is provided in Appendix E.

# 4 Related work

207 Recent work on slice-based Wasserstein distances mainly focused on improving their projection
208 efficiency, leading to a reduced number of projections needed to capture the structure of data
209 distributions [Kolouri et al., 2019a; Nguyen et al., 2021]. The GSWD proposes using nonlinear
210 projections to achieve this goal, and it has been proved to be a valid distance metric if and only if
211 they adopt injective GRTs, which only include the circular functions and a finite number of harmonic
212 polynomial functions with odd degrees as their feasible defining functions [Ehrenpreis, 2003]. While
213 the GSWD has shown impressive performance in various applications [Kolouri et al., 2019a], its
214 defining function is restricted to the aforementioned limited class of candidates. In addition, the
215 selection of defining function is usually task-dependent and needs domain knowledge, and the impact
216 on performance from different defining functions is still unclear.

217 To tackle those limitations, [Kolouri et al., 2019a] proposed the GSWD-NN, which *directly* takes
218 the outputs of a neural network as its projection results without using the standard Radon transform
219 or GRTs. However, this brings three side effects: 1) The number of projections, which equals the
220 number of nodes in the neural network's output layer, is fixed, thus new neural networks are needed
221 if one wants to change the number of projections. 2) There is no random projections involved in the
222 GSWD-NN, as the projection results are determined by the inputs and weights of the neural network. 3)
223 The GSWD-NN is a pseudo-metric since it uses a vanilla neural network, rather than Radon transform

224 or GRTs, as its push-forward operator. Therefore, the GSWD-NN does not fit into the theoretical
225 framework of GSWD and does not inherit its geometric properties.

226 Another notable variant of the SWD is the distributional sliced Wasserstein distance (DSWD) [Nguyen
227 et al., 2021]. By finding a distribution of projections that maximizes the expected distances over these
228 projections, the DSWD can slice distributions from multiple directions while having high projection
229 efficiency. Injective GRTs are also used to extend the DSWD to the distributional generalized sliced
230 Wasserstein distance (DGSWD) [Nguyen et al., 2021]. Experiment results show that the DSWD and
231 the DGSWD have superior performance in generative modelling tasks Nguyen et al. [2021]. However,
232 neither the DSWD nor the DGSWD have solved the problem with the GSWD, i.e. they are still not
233 able to produce nonlinear projections adaptively.

234 Our contribution differs from previous work in three ways: 1) The ASWD is data-adaptive, i.e. the hyper-
235 surfaces where high-dimensional distributions are projected onto can be learned from data. This implies
236 one does not need to specify a defining function from limited choices. 2) Unlike GSWD-NN, the ASWD
237 takes a novel direction to incorporate neural networks into the framework of sliced-based Wasserstein
238 distances while maintaining the properties of sliced Wasserstein distances. 3) Previous work on introduc-
239 ing nonlinear projections into Radon transform either is restricted to only a few candidates of defining
240 functions (GRTs) or breaks the framework of Radon transforms (neural networks in GSWD-NN), in
241 contrast, the spatial Radon transform provides a novel way of defining nonlinear Radon-type transforms.

## 5 Experiments

243 In this section, we describe the experiments that we have conducted to evaluate performance of the
244 proposed distance metric. The GSWD leads to the best performance in a sliced Wasserstein flow
245 problem reported in [Kolouri et al., 2019a] and the DSWD outperforms the compared methods in
246 the generative modeling task examined in [Nguyen et al., 2021] on CIFAR 10 [Krizhevsky, 2009],
247 CelebA [Liu et al., 2015], and MNIST [LeCun et al., 1998] datasets (Appendix H.2). Hence, we
248 compare performance of the ASWD with the state-of-the-art distance metrics in the same examples
249 and report results as below[1]. We provide additional experiment results in the appendices, including a
250 sliced Wasserstein autoencoder (SWAE) [Kolouri et al., 2019b] using the ASWD (Appendix I), image
251 color transferring (Appendix J) and sliced Wasserstein barycenters (Appendix K).

252 To examine the robustness of the ASWD, throughout the experiments, we adopt the injective network
253 architecture given in Equation (15) and set $\phi_\omega$ to be a single fully-connected layer neural network
254 whose output dimension equals its input dimension, with a ReLU layer as its activation function.

### 5.1 Sliced Wasserstein flows

256 We first consider the problem of evolving a source distribution $\mu$ to a target distribution $\nu$ by minimizing
257 Wasserstein distances between $\mu$ and $\nu$ in the sliced Wasserstein flow task reported in [Kolouri et al.,
258 2019a].

$$\partial_t \mu_t = -\nabla \text{SWD}(\mu_t, \nu), \qquad (17)$$

259 where $\mu_t$ refers to the updated source distribution at each iteration $t$. The SWD in Equation (17) can
260 be replaced by other sliced-Wasserstein distances to be evaluated. As in [Kolouri et al., 2019a], the
261 2-Wasserstein distance was used as the metric for evaluating performance of different distance metrics
262 in this task. The set of hyperparameter values used in this experiment can be found in Appendix F.1.

263 Without loss of generality, we initialize $\mu_0$ to be the standard normal distribution $\mathcal{N}(0, I)$. We repeat
264 each experiment 50 times and record the 2-Wasserstein distance between $\mu$ and $\nu$ at every iteration. In
265 Figure 2, we plot the 2-Wasserstein distances between the source and target distributions as a function
266 of the training epochs and the 8-Gaussian, the Knot, the Moon, and the Swiss roll distributions are
267 respective target distributions. For clarity, Figure 2 displays the experiment results from the 6 best
268 performing distance metrics, including the ASWD, the DSWD, the SWD, the GSWD-NN 1, which
269 directly generates projections through a one layer MLP, as well as the GSWD with the polynomial
270 of degree 3, circular defining functions, out of the 12 distance metrics we compared.

271 We observe from Figure 2 that the ASWD not only leads to smaller 2-Wasserstein distances, but also
272 converges faster by achieving better results with fewer iterations than the other methods in these four

---

[1]Code to reproduce experiment results is available at : https://bit.ly/2Y23wOz.

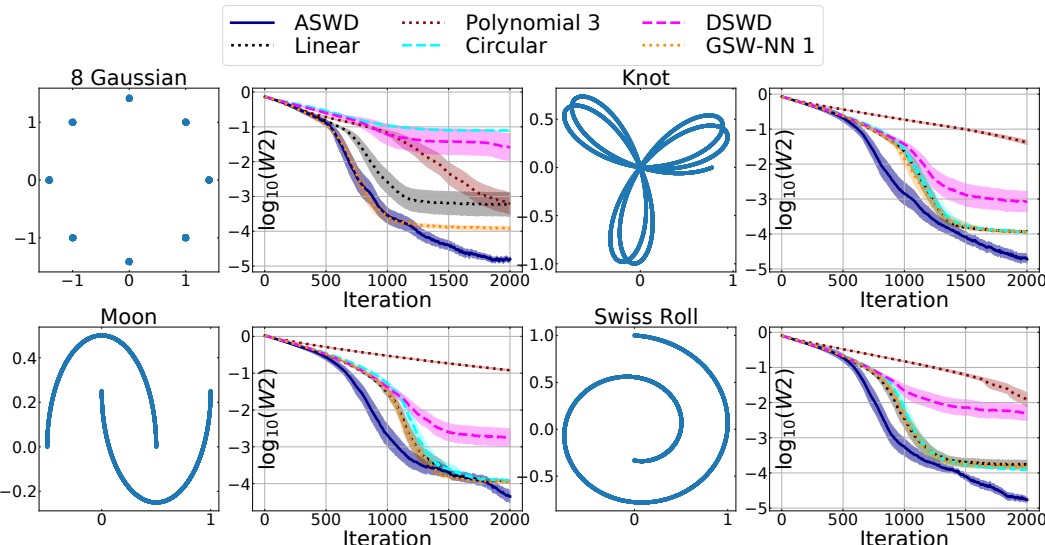

Figure 2: The first and third columns are target distributions. The second and fourth columns are log 2-Wasserstein distances between the target distribution and the source distribution. The horizontal axis show the number of training iterations. Solid lines and shaded areas represent the average values and 95% confidence intervals of log 2-Wasserstein distances over 50 runs. A more extensive set of experimental results can be found in Appendix G.1.

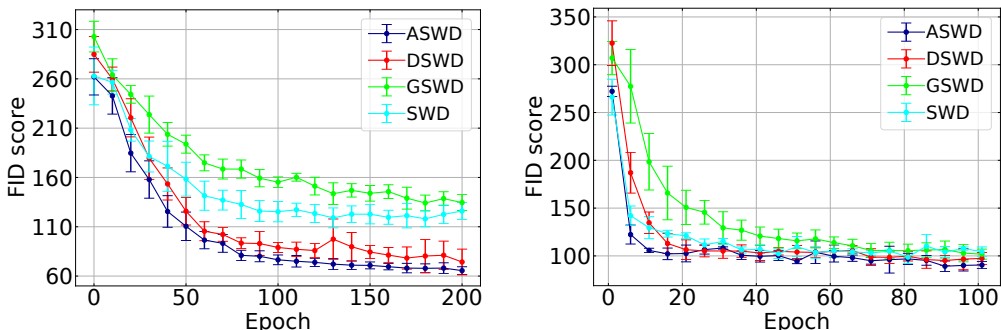

Figure 3: FID scores of generative models trained with different metrics on CIFAR10 (left) and CelebA (right) datasets with $L = 1000$ projections. The error bar represents the standard deviation of the FID scores at the specified training epoch among 10 simulation runs.

target distributions. A complete set of experimental results with 12 compared distance metrics and 8 target distributions are included in Appendix G.1. The ASWD outperforms the compared state-of-the-art sliced-based Wasserstein distance metrics with 7 out of the 8 target distributions except for the 25-Gaussian. This is achieved through the simple injective network architecture given in Equation (15) and a one layer fully-connected neural network with equal input and output dimensions throughout the experiments. In addition, ablation study is conducted to study the effect of injective neural networks, the optimization of hypersurfaces in the ASWD. Details can be found in Appendix G.2.

## 5.2 Generative modeling

In this experiment, we use the sliced-based Wasserstein distances for a generative modeling task described in [Nguyen et al., 2021]. The task is to generate images using generative adversarial networks (GANs) [Goodfellow et al., 2014] trained on either the CIFAR10 dataset ($64 \times 64$ resolution) [Krizhevsky, 2009] or the CelebA dataset ($64 \times 64$ resolution) [Liu et al., 2015]. Denote the hidden layer and the output layer of the discriminator by $h_\psi$ and $D_\Psi$, and the generator by $G_\Phi$, we train GAN

Table 1: FID scores of generative models trained with different distance metrics. Smaller scores indicate better image qualities. $L$ is the number of projections, we run each experiment 10 times and report the average values and standard errors of FID scores for CIFAR10 dataset and CELEBA dataset. The running time per training iteration for one batch containing 512 samples is computed based on a computer with an Intel (R) Xeon (R) Gold 5218 CPU 2.3 GHz and 16GB of RAM, and a RTX 6000 graphic card with 22GB memories.

| | SWD [Bonneel et al., 2015] | | GSWD [Kolouri et al., 2019a] | | DSWD [Nguyen et al., 2021] | | ASWD | |
|---|---|---|---|---|---|---|---|---|
| $L$ | FID | $t$ (s/it) | FID | $t$ (s/it) | FID | $t$ (s/it) | FID | $t$ (s/it) |
| | | | | CIFAR10 | | | | |
| 10 | 192.6±5.7 | 0.32 | 189.5±6.0 | 0.35 | 79.0±4.2 | 0.48 | **73.2±3.1** | 0.55 |
| 100 | 155.0±2.9 | 0.32 | 155.9±3.2 | 0.70 | 72.2±8.2 | 0.51 | **66.7±3.2** | 0.57 |
| 1000 | 126.0±2.9 | 0.34 | 134.5±2.7 | 2.10 | 74.3±4.3 | 1.22 | **65.5±3.9** | 1.32 |
| | | | | CELEBA | | | | |
| 10 | 118.3±3.1 | 0.32 | 143.2±5.5 | 0.35 | 105.3±3.4 | 0.49 | **99.2±4.3** | 0.53 |
| 100 | 116.0±2.8 | 0.33 | 120.8±1.8 | 0.69 | 103.1±3.8 | 0.51 | **94.3±2.2** | 0.56 |
| 1000 | 104.4±2.8 | 0.34 | 101.8±1.8 | 2.14 | 97.4±2.1 | 1.21 | **90.5±3.0** | 1.31 |

models with the following objectives:

$$\min_{\Phi} \text{SWD}(h_\psi(p_r), h_\psi(G_\Phi(p_z))), \tag{18}$$

$$\max_{\Psi,\psi} \mathbb{E}_{x \sim p_r}[\log(D_\Psi(h_\psi(x)))] + \mathbb{E}_{z \sim p_z}[\log(1 - D_\Psi(h_\psi(G_\Phi(z))))], \tag{19}$$

where $p_z$ is the prior of latent variable $z$ and $p_r$ is the distribution of real data. The SWD in Equation (18) is replaced by the ASWD and other variants of the SWD to compare their performance. The GSWD with the polynomial defining function and the DGSWD is not included in this experiment due to its excessively high computational cost in high-dimensional space. The *Fréchet Inception Distance* (FID score) [Heusel et al., 2017] is used to assess the quality of generated images. More details on the network structures and the parameter setup used in this experiment are available in Appendix F.2.

We run 200 and 100 training epochs to train the GAN models on the CIFAR10 and the CelebA dataset, respectively. Each experiment is repeated for 10 times. We report experimental results in Table 1. With the same number of projections and a similar computation cost, the ASWD leads to significantly improved FID scores among all evaluated distances metrics on both datasets, which implies that images generated with the ASWD are of higher qualities. Figure 3 plots the FID scores recorded during the training process. The GAN model trained with the ASWD exhibits a faster convergence as it reaches smaller FID scores with fewer epochs. Randomly selected samples of generated images are presented in Appendix H.1.

# 6 Conclusion

We proposed a novel variant of the sliced Wasserstein distance, namely the augmented sliced Wasserstein distance (ASWD), which is flexible, has a high projection efficiency, and generalizes well. The ASWD adaptively updates the hypersurfaces used to slice compared distributions by learning from data. We proved that the ASWD is a valid distance metric and presented its numerical implementation. We reported empirical performance of the ASWD over state-of-the-art sliced Wasserstein metrics in various numerical experiments. We showed that ASWD with a simple injective neural network architecture can lead to the smallest distance errors over the majority of datasets in a sliced Wasserstein flow task and superior performance in generative modeling tasks involving GANs and VAEs. We have also evaluated the applications of the ASWD in downstream tasks including color transferring and Wasserstein barycenters. What remains to be explored is the impact of the injective neural network architecture used in the ASWD, e.g. the application of different types of invertible neural networks in the ASWD framework. We leave this topic as a potential research direction of our future work.

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
