# OpenReview forum: "Augmented Sliced Wasserstein Distances"
_NeurIPS.cc/2021/Conference — NeurIPS 2021 Submitted_

### Official Review · Reviewer_xvVx · 2021-06-26

**Rating:** 7
**Confidence:** 4

**Summary:**

The paper develops a computationally efficient metric between two probability distributions, which is called Augmented Sliced Wasserstein Distances (ASWD). The proposed metric applied neural networks to map two distributions from $\mathbb{R}^d$ into high dimensional non-linear manifolds, then the mapped distributions are compared using the sliced Wasserstein distance. It is shown in the paper that ASWD is a well-defined metric, and an algorithm for empirically evaluating ASWD is provided. The empirical applicability of ASWD is demonstrated by numerical experiments including gradient flow, generative modeling, sliced Wasserstein autoencoder, image color transferring and
sliced Wasserstein barycenter.

**Limitations And Societal Impact:**

The author discussed societal impact in appendix L. An additional limitation of this work could be:

- As ASWD applied neural networks to map the data, the resulting metrics might be hard to interpret.

**Main Review:**

In general the paper is well-written, the proof is rigorous, and the numerical result is convincing.

**major concerns:**

1. How to select the regularization parameter $\lambda$ is not discussed in the paper. Though it is discussed in Remark 5 that $\lambda$ adjusts the introduced non-linearity and $\lambda$ can not be too small (the output $g_{\omega}$ might explode) or too large (totally kills the nonlinear term), how to select the lambda in practice is not clear to the readers. In appendix F, the authors report: empirical errors in the experiment are found to be not sensitive to the choice of $\lambda$ in a candidate set of {0.01, 0.05, 0.1, 0.5}: does it means that the candidate set is not rich enough so that one can not observe variations?

2. Empirically, the neural network $g$ is selected as $g^\ast = argmax(\mathcal L(\mu,\nu;g,\lambda))$ instead of $g^\ast = argmax(\mathrm{ASWD}(\mu,\nu;g))$. With such $g^\ast$, is the result in Corollary 1.1 still valid?

**Minor suggestions:**

1. Line 81: Is it necessary to assume the existence of PDFs for $\mu$ and $\nu$? Many interesting data-driven applications does not satisfy this assumptions. For instance, the empirical measure of a finite observations is not absolutely continuous w.r.t Lebesgue measure.

2. Line 94: $\delta(\cdot)$ is the **1-dimensional** Dirac delta function.

3. Line 434: need to restrict to measurable $\mathcal X$, otherwise the integral is not well defined.

**Time Spent Reviewing:**

8

---

> ### Author Response · Authors · 2021-08-10
> **Response to Reviewer xvVx**
>
> Many thanks for your positive and constructive feedback! Please see below our response to your comments:
>
> (1) **How to select the regularization parameter $\lambda$ is not discussed in the paper ... does it means that the candidate set is not rich enough so that one can not observe variations?**
>
> **[Select the regularization parameter]** As we reported in Appendix F, the value of $\lambda$ is selected from a candidate set. Cross-validation was used to find a favourable value for $\lambda$. We will clarify this in the revised paper.
>
> **[The candidate set is not rich enough]** We have now evaluated the performance of the ASWD when the range of $\lambda$ is much larger. Specifically, when $\lambda=10$ or $100$, the resulted ASWD leads to decreased performance on par with the SWD in the sliced Wasserstein flow example. This is consistent with our expectation that excessive regularization will eliminate nonlinearity as discussed in Remark 5, leading to similar performance with the SWD. We will add the discussion after the rebuttal.
>
> (2) **Empirically, the neural network $g$ is selected as $g^\*=$argmax $\mathcal{L}(\mu, \nu; g, \lambda)$ instead of $g^\*=$argmax $\mathcal{L}(\mu, \nu; g)$. With such $g^\*$, is the result in Corollary 1.1 still valid?**
>
> With the optimized mapping $g^\*=$**argmax** $\mathcal{L}(\mu, \nu; g, \lambda)$, the Corollary 1.1 is still valid. In Corollary 1.1, the optimal mapping $g^\*$ is defined as **argmax** $\mathcal{L}(\mu, \nu; g)$ because the mapping $g(\cdot)$ is already confined to the sets of bounded and injective functions. While in our numerical implementation, the regularization term is introduced to empirically restrict the optimization in the above desired set, resulting in the mapping $g^\*=$**argmax** $\mathcal{L}(\mu, \nu; g, \lambda)$, thus $g^*=$**argmax** $\mathcal{L}(\mu, \nu; g, \lambda)$ is just a realization of the optimal mapping in Corollary 1.1. We will add the discussion in the revision.
>
> (3) **Is it necessary to assume the existence of PDFs for $\mu$ and $\nu$?**
>
> We acknowledge that the assumption is not necessary, and rephrase it using a measure-based formalism.
>
> (4) **Minor suggestions: Line 94: $\delta(\cdot)$ is the 1-dimensional Dirac delta function. Line 434: need to restrict to measurable $\mathcal{X}$, otherwise the integral is not well defined.**
>
> We will revise the paper accordingly.

---

> > ### Comment · Reviewer_xvVx · 2021-08-19
> > **Response to Paper2171 Authors**
> >
> > I would like to thank the authors to provide detailed explanations to all my concerns and questions. As I received satisfactory responses to my questions, I will change my evaluation accordingly.

---

### Official Review · Reviewer_7Tc3 · 2021-07-07

**Rating:** 6
**Confidence:** 4

**Summary:**

The author proposed a variant of sliced Wasserstein distance (SWD) called **augmented sliced Wasserstein distance** (ASWD) which replace the linear projections in SWD with a non-linear one. The construction of ASWD consists of two steps: (1) define spatial Radon transform (SRT) for probability measures (2) replace the Radon transform in the original SWD with this new SRT. The key intuition of SRT is to first transform the function $\pmb{x}$ to high dimensional hyperspace by transformation $g(\pmb{x})$ and then perform Radon transform on functions defined in this hyperspace.

Theoretically, the author proves the conditions required for SRT to be injective. Later, the author uses this result to show the corresponding ASWD is a distance metric. In addition, the author proposes a training objective for finding the transformation $g(\cdot)$.

Empirically, the author conducts both toy and practical experiments to demonstrate the advantages of ASWD in terms of higher projection efficiency and also provides some ablation studies in the appendix to investigate the effects of different $g(\cdot)$ choices.

**Limitations And Societal Impact:**

For limitations, see Main Review.

Appendix L (societal impacts) addresses the potential negative social impact of this work.

**Main Review:**

### Clarity
The main idea of ASWD is easy to understand and the author also gives a quite clear explanation on how to construct ASWD. Specifically, I appreciate the **Remarks**, which greatly improves the clarity of the paper. So I think the paper is clearly written and quite easy to follow.

### Originality
As for the originality, the idea of using injective transformation to preserve some statistical properties is not new. For example, this is the foundation of normalizing flow, and it can also be found in the deep-kernel literature [1] to preserve the discrepancy validity. However, this seems to be new in the scope of SWD literature. In summary, I found the originality is ok but not significantly novel.

### Quality and Significance
In terms of the research quality, the method is well-presented. I checked some of the proofs, they seem to be correct. Empirically, the author conducts several experiments to confirm the advantages of ASWD. However, I still have some concerns related to the experiments and the way the author presented the ASWD. In terms of significance, this work should be easy to implement and understand. So it should have some impact on the SWD community.

### Questions
1. You mentioned that the idea of SRT is to transform the original data $\pmb{x}\in\mathbb{R}^d$ to $\mathbb{R}^{d_\theta}$ and $d_\theta> d$, i.e. higher-dimensional space. However, I don't think this is necessary based on the theory you give. For structured data like images, the commonly used assumption is that they have a low-dimensional manifold. So I don't understand why you project it in higher-dimensional space. To me, it seems that this is due to the choice of the NN architecture (concatenation) and not from the theoretical arguments.
2. The form of the transformation $g$ is quite simple. Have you evaluated the performance of using the invertible neural network for $g$? Despite the computational cost, does it give better performance?
3. I wonder about the performance if you transform the data to low dimensional space by an arbitrary neural network. Indeed, strict injectivity cannot be guaranteed but one can roughly construct an injective network using the technique mentioned in [1].
4. What advantages of this strict injectivity can provide in practice? In the ablation studies, it seems that the non-injective NN can sometimes give even better performance compared to injective ASWD? Any real applications that actually require this strict injectivity?
5. Despite the ASWD gives better performance compared to other baselines, the FID score is still quite low compared to the current state-of-the-art method. So what real advantages of this SWD can bring?

Minor:
1. Is SRT a special case of GRT in general? If so, I wonder about the expressiveness of SRT. Any transformations that SRT cannot represents but GRT can?



[1] Li, Chun-Liang, et al. "Mmd gan: Towards deeper understanding of moment matching network." arXiv preprint arXiv:1705.08584 (2017).

---
I appreciate the author's response. They managed to address most of my concerns. However, I agree with reviewer HUCr about the gap between the theory and the definition of $\pmb{g}$. The search space of $\pmb{g}$ and the optimal $\pmb{g}^*$ should be re-defined to address this. Anyway, I agree that this work should have some potentials, I will raise my score.

**Time Spent Reviewing:**

4

---

> ### Author Response · Authors · 2021-08-10
> **Response to Reviewer 7Tc3**
>
> Many thanks for your constructive and insightful comments! Please see below our detailed response to your comments and we hope that it will resolve any concern you had:
>
> (1) **The idea of SRT is to transform the original data to higher dimensional space ... it seems that this is due to the choice of the NN architecture (concatenation) and not from the theoretical arguments.**
>
> We want to clarify that the SRT is not necessarily defined as transforming data to higher dimensional spaces. We mentioned ``higher-dimensional" in line 8 as the numerical implementation provided we adopted projects data to higher-dimensional space. We will revise this after rebuttal to avoid any confusion. Indeed, injections can be constructed as mappings projecting data to lower-dimensional spaces, but in practice it is non-trivial to construct such injections without further knowledge about the data structure.
>
> (2) **The form of the transformation is quite simple. Have you evaluated the performance of using the invertible neural network for $g$? Despite the computational cost, does it give better performance?**
>
> We have evaluated the performance of the ASWD with invertible neural networks such as the RealNVP, planar flow, and radial flow [1] [2] in the slice Wasserstein flow experiment. Specifically, the numerical results we obtained show that the ASWD defined with RealNVP, planar flow and radial flow produced better performance than GSWD variants in most setups. They exhibit slightly worse performance compared with the ASWD with injective mapping defined in Equation (15), possibly due to the additional restriction in invertible mapping imposed by the RealNVP, planar flow and radial flow. We will discuss the performance of ASWD with these invertible neural network architecture in the revision.
>
> (3) **I wonder about the performance if you transform the data to low dimensional space by an arbitrary neural network. Indeed, strict injectivity cannot be guaranteed but one can roughly construct an injective network using the technique mentioned in [3].**
>
> The performance of arbitrary non-injective neural networks with the same dimensional outputs, denoted as ASWD-vanilla-non-injective, has been reported in the ablation study in Appendix G.2 which showed that it led to much worse performance than the ASWD. Thanks for your suggestion of the candidate injective functions in [3]. We will conduct the suggested experiments and add the results after the rebuttal.
>
> (4) **What advantages of this strict injectivity can provide in practice? In the ablation studies, it seems that the non-injective NN can sometimes give even better performance compared to injective ASWD? Any real applications that actually require this strict injectivity?**
>
> **[Advantages of strict injectivity in practice; non-injective NN can sometimes give better performance]** In fact, both the ASWD-non-injective and the ASWD-vanilla-non-injective in Section G.2 adopt non-injective neural networks whose input and output have the same dimensionality, but their relative performance is significantly different when compared with their injective counterparts, the ASWD and the ASWD-vanilla. Specifically, the performance of the ASWD-vanilla-non-injective is worse than its counterpart, ASWD-vanilla, implying that the theoretical properties of injective networks can help distinguish the compared distributions. On the other hand, the ASWD-non-injective has similar performance as the ASWD, and in some datasets it outperforms its injective counterpart in this ablation study at the cost of more unstable training. Among all datasets in this ablation study, we observe that the ASWD with injective mapping consistently lead to more stable training compared to that with non-injective mapping and outperforms SWD/GSWD.
>
> **[Applications that actually require strict injectivity]** The injectivity is used here to guarantee the metric property of the resulted ASWD. The metric property, including the triangle inequality, can be useful in applications such as similarity searching problems to reduce the search space -- with triangle inequality, we have $d(u,v) \geq d(u,x) -  d(v,x)$ for any three points $u, v, x$ in a space with the distance metric $d$. Thus, if we know the values of $d(u,x)$ and $d(v,x)$, we can use them to compute the lower bound on $d(u,v)$ after a simple subtraction. This can be useful to prune off a significant fraction of the distance computations in such applications.
>
>
>
> (5) **Despite the ASWD gives better performance compared to other baselines, the FID score is still quite low compared to the current state-of-the-art method. So what real advantages of this SWD can bring?**
>
>  The generative modelling experiment is conducted to compare the ASWD with other state-of-the-art sliced-based Wasserstein metrics (DSWD in this example) in the same experiment setup where the DSWD has been evaluated and reached favorable performance. As reported in the experiment results, the ASWD outperforms the other compared slice-based Wasserstein metrics in almost all setups. Comparison with other generators that are unrelated with slice-based Wasserstein metrics is a topic worth further investigation, but we feel that this is out of the scope of this paper.
>
> (6) **Is SRT a special case of GRT in general? If so, I wonder about the expressiveness of SRT. Any transformations that SRT cannot represents but GRT can?**
>
> We would like to clarify that SRT is not a special case of GRT. In particular, there are only two families of defining functions that GRTs can employ, i.e. the circular and the polynomial defining functions [4]. The circular and polynomial defining functionss do not include the class of defining functions of SRTs as SRTs can be constructed using injective neural networks and have flexible functional forms, while the the functional forms of the circular and polynomial defining functions are fixed. On the other hand, the SRT with injective neural networks cannot represent the GRT defined with the circular defining function. However, the SRTs can represent more flexible transformations than the GRT in general, and the performance of the GSWD defined with the GRT is shown to lead to poorer performance than the ASWD in almost all experiment setups evaluated in the paper across a diverse problem domains.
>
> [1] L. Dinh, J. Sohl-Dickstein, and S. Bengio.  Density estimation using real NVP. arXiv:1605.08803, 2016.
>
> [2] D. Rezende and S. Mohamed. Variational inference with normalizing flows. In Proc. International Conference on Machine Learning (ICML), pp. 1530–1538, Lille, France, 2015.
>
> [3] C. Li, W. Chang, Y. Cheng, Y. Yang, and B. Poczos. MMD gan: Towards deeper understanding of moment matching network. arXiv:1705.08584, 2017.
>
> [4] S. Kolouri, K. Nadjahi, U. Simsekli, R. Badeau, and G. Rohde. Generalized sliced Wasserstein distances. In Proc. Advances in Neural Information Processing Systems (NeurIPS), pp. 261–272, Vancouver, Canada, 2019.

---

> ### Author Response · Authors · 2021-09-01
> **Thank you for your comment**
>
> Thank you very much for your feedback and raising your score. We are delighted to know that we have addressed most of your concerns. We would also like to update with you that we have now re-defined $g^*$ based on the suggestion from Reviewer HUCr as in "Update on Corollary 1.1" and the following discussions in the response to Reviewer HUCr. We hope that this update and the accompanied proof have now addressed your remaining concern of the gap between the theory and the definition of $g$.

---

### Official Review · Reviewer_boZk · 2021-07-12

**Rating:** 5
**Confidence:** 4

**Summary:**

The sliced Wasserstein distance (SWD) is a metric on the space of probability measures and serves as a practical alternative to the standard metric arising from the optimal transport (OT) problem, the Wasserstein distance. Indeed, while the Wasserstein distance suffers from computational and statistical limitations on large-scale and high-dimensional settings, SWD can alleviate these issues while offering similar theoretical properties. Hence, SWD has been increasingly popular within the statistical and machine learning (ML) community in recent years.

SWD is defined as an expectation that is intractable in general; in practice, it is then usually approximated with a simple Monte Carlo estimation, but prior studies reported that this method might induce an important approximation error on high-dimensional settings, leading to an underestimation of the "true" dissimilarity between two distributions. Therefore, one would need to compute a Monte Carlo approximation based a very large amount of samples to obtain an accurate approximation, which increases the computational complexity of SWD.

Several variants of SWD have recently been introduced to overcome this problem and include the "generalized sliced Wasserstein distances" (GSWD): this class of distances aim at offering a better computational efficiency, by comparing distributions from their nonlinear one-dimensional representations (instead of the linear ones used in SWD). This paper focuses on this line of work: the limitations of GSWD are identified to motivate the formulation of a novel class of distances, called "Augmented Sliced Wasserstein distances" (ASWD).

Similarly to SWD and GSWD, the definition of ASWD is connected to the Radon transform: the authors introduce a novel type of Radon transform, the "spatial Radon transform" (Definition 1), and use it to define the class of augmented sliced Wasserstein distances.
Then, they study the theoretical properties of the spatial Radon transform and ASWD. Their main result states that ASWD verify all metric axioms if and only if the mapping $g$ (that characterizes the underlying spatial Radon transform) (1) is injective (Theorem 1), or (2) maximizes ASWD over the space of bounded and injective functions (Corollary 1.1).
The authors then introduce a specific class of mappings (Equation (15)) which meet the conditions of Corollary 1.1 thus guarantee that the resulting ASWD is a metric. Such mappings require training a neural network to make ASWD data-adaptive.
Finally, the superior empirical performance of ASWD against existing variants of SWD (including GSWD) is illustrated on various problems, including gradient flows, generative modeling, color transferring and barycenters of measures.


**Ethical Concerns:**

I don't have any ethical concerns for this paper.

**Limitations And Societal Impact:**

The limitations of this work are not clearly discussed, but the authors adequately addressed the societal impact (see Appendix L).


**Main Review:**

This work proposes a new family of distances inspired by computational optimal transport, which can be easily applied to various classical machine learning applications. In that sense, it fits the interests of the NeurIPS community.

I found the paper quite clear and well-written overall. The structure can be improved by incorporating the following changes,
- l.46-47: "Different variants of the SWD have been proposed to improve the projection efficiency of the SWD, either by introducing nonlinear projections or by optimizing the distribution of random projections." It seems more relevant to talk about the max-sliced Wasserstein distance (max-SWD) in this paragraph instead of in the previous one, since max-SWD was introduced in order to improve the "projection complexity issue" of SWD (see "Max-Sliced Wasserstein Distance and its use for GANs", Deshpande et al., 2019).
- I suggest adding the equation that explains how to approximate equation (2) at the end of the paragraph "Wasserstein distance" (Section 2). This would clarify l.113-115, as well as equations (9) and (10)
- l.101-105: I would remove this part since explaining how to approximate the Radon transform or GRT does not seem useful in this work. On the contrary, specifying the approximation for the regularization term in Section 3.2 (l.198) would be more relevant.
- l.168: "In practical applications, the mapping $g(\cdot)$ needs to be optimized." At this stage, it is not obvious why this optimization is required. I suggest moving the explanation in Section 3.2 (e.g., l.191-196) to clarify this point.

Overall, the theoretical claims look correct to me; my only concerns are detailed below.
- Remark 2: "$g(\cdot)$ is a homogeneous polynomial function with odd degrees" (l.151) contradicts equation (31) in Appendix B. On the other hand, it is not clear to me why the mapping defined in equation (31) is injective on $\mathbb{R}^d$.
- Appendix D: Why is it important to assume that the optimal mapping is bounded? Besides, equations (39), (40), (41) could be explained in more details.

Other (minor) typos and inaccuracies are listed at the end of the review.

In my opinion, the main weakness of this work, which prevents me from giving a positive score, is the lack of originality and significance as compared to prior work. Indeed, ASWD build on GSWD and their definition differs from a simple trick (using the "spatial Radon transform" introduced by the authors, instead of the generalized Radon transform). The structure of this paper is the same as in "Generalized Sliced Wasserstein Distances" (Kolouri et al., 2019) [1]: the novel distance is defined, then its metric axioms are studied, finally its performance against the sliced Wasserstein distance (and its variants) is evaluated on practical applications. These two aspects are not necessarily concerning and help make the paper quite intuitive; besides, the empirical study provided by this paper is more extensive than the one in [1], since the compared methods and applications are more diverse. What is more questionable is that the advantages of ASWD as compared to SWD and GSWD are unclear to me in the end, given the contributions and discussion provided in this work.

First, from a theoretical point of view, this paper only investigates the metric properties, and establishes that ASWD satisfy all axioms of metrics if and only if the mapping associated to its Radon transform is injective. While the authors argue that this is a significant advantage over GSWD, we don't know if this explains the superior performance of ASWD-based methods in practice. Besides, this result is similar to Proposition 1 in [1], which states that GSWD are metrics if and only if the corresponding generalized Radon transform is injective; consequently, the proofs in the appendix are largely inspired by [1].

Then, the empirical performance of ASWD is not sufficiently discussed: while the results in Section 5 encourage its use against existing slice-based distances, the appendix illustrates that ASWD do not necessarily yield a superior performance:
- Appendix I and K: ASWD and SWD achieve a similar performance, while ASWD is more computationally expensive.
- Appendix I and J: the performance of GSWD should be studied on these experiments as well.

This shows that the superior performance of ASWD depends on the application, and the theoretical results do not help understand the behavior in practice. In particular, I am wondering whether the regularization term (equation (16)) might help improve the performance in the flows and generative modeling applications: it would be interesting to regularize the other methods to study the impact.

Hence, I encourage the authors to provide more significant evidence on why ASWD is a better alternative to SWD than GSWD, or to show a more nuanced picture in light of the empirical results in the appendix; for now, I am not convinced by the superiority of ASWD over SWD/GSWD given the motivation (in particular, l.234-241) nor the theoretical and empirical contributions.

Typos:

Equation (1): the integral should be over $\Omega \times \Omega$

l.92: "over spaces of lines in $\mathbb{R}^d$" -> "over spaces of hyperplanes in $\mathbb{R}^d$"

l.99: "while $\Omega_\theta$ is a compact set" -> "where $\Omega_\theta$ is a compact set"

l.110: "where the Radon transform ... is adopted as the measure push-forward operator" This can be further clarified by explaining the connection between the Radon transform and the push-forward operator that projects the two distributions (Proposition 6 in "Sliced and Radon Wasserstein Barycenters of Measures", Bonneel et al.).

l.120: "The GSWD is a valid metric except for its neural network variant [Kolouri et al., 2019a]" This is not exactly true: it remains unknown whether neural networks-based GSWD (GSWD-NN) satisfy all axioms for a metric (see the explanation in Section 3.3 of "Generalized Sliced Wasserstein Distances", Kolouri et al.). Therefore, a correct statement would be, for example, "The GSWD has been shown to be a valid metric, but the proof remains incomplete for its neural network variant [Kolouri et al., 2019a]".

Definition 1:
- The set $P(\mathbb{R}^d)$ is not defined
- What are the possible values for $d$ and $d_\theta$?

l.147: please define the operator "\equiv".

l.172: "between two probability" -> "between two probability measures"

Section 5: The order $k$ of SWD/ASWD used in the experiments should be precised.

l.256-257: "by minimizing Wasserstein distances" -> "by minimizing slice-based Wasserstein distances"

l.425, 458, 463: "for $\forall x \in \mathbb{R}^d$" -> "for all $x \in \mathbb{R}^d$". Same remark in l.428, 432, 434, 506...

l.455: "which probability density function is ..." -> "whose probability density function is ..."

l.498: "that the the ASWD"

l.589: "Sampels of generated images" -> "Samples of generated images"


**Time Spent Reviewing:**

8

---

> ### Author Response · Authors · 2021-08-10
> **Response to Reviewer boZk (1/2)**
>
> Many thanks for your constructive and insightful feedback! Please see below for our detailed response to your comments which we hope would resolve any concerns you had:
>
> (1) **I found the paper quite clear and well-written overall. The structure can be improved by incorporating the following changes.**
>
> We will incorporate your comments to improve the structure of the paper.
>
> (2) **Remark 2: "$g(\cdot)$ is a homogeneous polynomial function with odd degrees" (l.151) contradicts equation (31) in Appendix B.**
>
> We will revise Remark 2 as follows.
>
> The spatial Radon transform degenerates to the vanilla Radon transform when the mapping $g(\cdot)$ is an identity mapping. The spatial Radon transform is equivalent to the polynomial GRT [1] when $g(x)=(x^{\alpha_1},...,x^{\alpha_{d_\alpha}})$, where $\alpha$ is multi-indices $\alpha_i=(\eta_{i,1},...,\eta_{i,d})\in \mathbb{N}^{d}$ satisfying $|\alpha_i|=\sum_{j=1}^{d} \eta_{i,j}=m$. $m$ is the degree of the polynomial function, $x^{\alpha_i}=\prod_{j=1}^{d} x_j^{\eta_{i,j}}$ given an input $x=(x_1,...,x_d)\in \mathbb{R}^d$, and $d_\alpha$ is the number of all possible multi-indices $\alpha_i$ that satisfies $|\alpha_i|=m$.
>
> (3) **It is not clear to me why the mapping defined in equation (31) is injective on $\mathbb{R}^d$.**
>
> We illustrate this with an example in the 2-dimensional space. When $m=3$ in Equation (30) and $x=(x_1, x_2)$, the output of $g(x)=(x^{\alpha_1}, \cdots, x^{\alpha_{d_\alpha}})$ as defined in Equation (31) is a vector $g(x)=(x_1^3, x_1^2 x_2, x_1 x_2^2, x_2^3)$, which is clearly an injection as the elements $x_1^3$ and $x_2^3$ are uniquely determined by the input $(x_1,x_2)$. This can be straightforwardly generalized to higher-dimensional spaces with odd degrees. When $x=(x_1, \cdots ,x_d)$ and the degree $m$ is odd, the elements $x_1^m, x_2^m, \cdots ,x_d^m$ in the output of $g(x)$ are again uniquely determined by the input $x$. Therefore, the mapping $g(x)$ defined in Equation (31) is an injection.
>
> (4) **Why is it important to assume that the optimal mapping is bounded? Besides, equations (39), (40), (41) could be explained in more details.**
>
> The mapping needs to be bounded because otherwise the output of $g(\cdot)$ can be arbitrarily large when optimizing it and as a result the ASWD$(\mu, \nu; g)$ can be unbounded. (39) is from the triangle inequality of the ASWD defined with a fixed mapping function ($g_1(\cdot)$ in this case and proved for Theorem 1); (40) can be obtained by expanding the $k$-th power of r.h.s of (39) and the non-negativity of the ASWD defined with a fixed mapping function; (41) can be directly derived from the definition of supremum. We will add the clarification in the revised version of the paper.
>
> (5) **The main weakness of this work, which prevents me from giving a positive score, is the lack of originality and significance as compared to prior work ... The advantages of ASWD as compared to SWD and GSWD are unclear to me in the end, given the contributions and discussion provided in this work.**
>
> **[Originality and significance]** We would like to highlight here the advantages and significance of the ASWD compared to the SWD and GSWD. The SWD and GSWD are seminal works and as Reviewer HUCr pointed out ``This paper heavily builds on previous works (in particular those of [1]) which it properly credits. It provides simple yet seemingly efficient variations that improve on previous results.'' As stated in line 234-241 in the paper, the significance of the work is that the proposed ASWD overcomes a major limitation of previous efforts including the GSWD in introducing nonlinearity to Radon transform (and slice-based Wasserstein distances), i.e. the choice of the defining functions, in the state-of-the-art methods such as the GSWD, are limited, problem specific, and requires domain knowledge/experiments to hand pick the most suitable defining functions for a particular application. In contrast, the proposed ASWD is data-adaptive -- the hypersurfaces where high-dimensional distributions are projected onto can be end-to-end optimized with minimal tuning required. In fact, we adopt a simple injective network structure to construct the ASWD throughout the large set of experiments in diverse domains (as stated in line 252-254), which resulted in a clear advantage in its robust empirical performance compared with the SWD and the GSWD -- the ASWD led to the best performance in 15 out of 17 sub-experiments we evaluated where quantitative results are presented. More details on its superior empirical performance are elaborated below.
>
> **[The advantages of the ASWD compared to the SWD and the GSWD]** As Reviewer HUCr pointed out, "the paper is overall interesting and supported by a large number of convincing experiments", in areas of "gradient flow, generative models, sliced Wasserstein auto-encoder, sliced Wasserstein barycenter, and color transferring". These experiments consist of 26 sub-experiments, 17 of them were reported with quantitative results, and 9 of them were presented with visualized qualitative results. Specifically, we observe that the ASWD led to the best performance in **15 out of 17** experiment setups in gradient flow, generative models, and sliced Wasserstein auto-encoder experiments, where quantitative metrics were used to evaluated performance of compared methods. In comparison, any particular form of the GSWD or DSWD (with a particular defining function) achieved the best performance in 1 out of these 17 experiment setups, and the GSWD or DWSD (with any choice of defining functions) achieved the best performance in 2 out of 17 experiment setups. It is worth noting that the gradient flow example in Section 5.1 adopts the same experiment setup where the GSWD reported its superior performance [1] and the generative modeling example in Section 5.2 uses the same experiment setup from the DSWD work [2]. The results quantitatively demonstrate the advantage of the ASWD that it is able to capture the most discriminating projection hypersurface, thus leading to higher projection efficiency than the compared slice-based Wasserstein metrics in diverse problem settings. Besides, in qualitative experiments such as sliced Wasserstein barycenter and color transferring, the ASWD produces visually smooth and realistic images and meaningful barycenters.
>
> Details in the aforementioned 17 quantitative experiment setups and results are listed as follows. In the gradient flow experiment, 8 different datasets are included and the ASWD achieved the best performance in 7 datasets. The generative modelling experiment includes 8 different experiment setups where the CIFAR10, the CelebA, the MNIST datasets are employed. Specifically, the ASWD led to the best FID scores among all the compared methods with 10, 100, and 1000 projections on the CIFAR10 and the CelebA datasets, and the ASWD also outperformed all the other methods regarding both the 2-Wasserstein distance and the SWD between generated and real images on the MNIST datasets with 10 and 1000 projections. In the Sliced Wasserstein auto-encoder experiment, the evaluated approaches were tested on the MNIST dataset and the ASWD produced lower 2-Wasserstein distances  between the encoded latent space and the prior distribution compared to the SWD although these two methods led to similar cross-entropy loss.
>
> (6) **First, from a theoretical point of view, this paper only investigates the metric properties, ... Besides, this result is similar to Proposition 1 in [1], ... consequently, the proofs in the appendix are largely inspired by [1].**
>
> As we discussed in (5), the theoretical motivation and advantage of this work include that it addresses a major issue in introducing nonlinearity in the slice-based Wasserstein distances, i.e. the higher projection efficiency in state-of-the-art methods such as the GSWD often comes at the cost of limited, problem-specific defining functions that require domain knowledge (and trial and error) in its application in real-world applications. As stated in line 238-241, if the GSWD [1] incorporates more flexible nonlinear projections using neural networks, the resulted GSWD-NN breaks the framework of Radon transforms and loses its metric property shown in Proposition 1 in [1]. Hence, it is significant to design the proposed ASWD and validate theoretically that it can incorporates highly flexible injective neural network structure while maintaining its metric property, something not achieved in previous works.
>
> The superior empirical performance of ASWD-based methods can be explained by the fact that the design of the ASWD matches well with its theoretical motivation, that it is data-adaptive, robust and highly effective across diverse problem settings with minimal tuning required.
>
> [1] S. Kolouri, K. Nadjahi, U. Simsekli, R. Badeau, and G. Rohde. Generalized sliced Wasserstein distances. In Proc. Advances in Neural Information Processing Systems (NeurIPS), pp. 261–272, Vancouver, Canada, 2019.
>
> [2] K. Nguyen, N. Ho, T. Pham, and H. Bui. Distributional sliced-Wasserstein and applications to generative modeling. In Proc. International Conference on Learning Representations (ICLR), Vienna, Austria, 2021.

---

> > ### Author Response · Authors · 2021-08-10
> > **Response to Reviewer boZk (2/2)**
> >
> > (7) **The empirical performance of ASWD is not sufficiently discussed ... Appendix I and K: ASWD and SWD achieve a similar performance, while ASWD is more computationally expensive. Appendix I and J: the performance of GSWD should be studied on these experiments as well.**
> >
> > **[Appendix I and K: ASWD and SWD achieve a similar performance]** We want to point out that the ASWD outperforms the SWD in almost all experiment setups as we summarised in the Response point (5). Even in Appendix I, the ASWD also produces latent distributions closer to the prior distribution in terms of 2-Wasserstein distances (Figure 10 (b)), which implies the ASWD can provide more informative projections than the SWD. In Appendix K, there is no quantitative metric we can employ to evaluate the quality of generated barycenters, but barycenters generated by the ASWD are visually more meaningful than those generated by the GSWD and the DSWD as shown in Figure 14.
> >
> > **[Appendix I and J: The performance of GSWD should be studied]** Thanks for the suggestion. We will include the GSWD in these experiments in the revision after the rebuttal.
> >
> > (8)  **This shows that the superior performance of ASWD depends on the application, and the theoretical results do not help understand the behavior in practice. In particular, I am wondering whether the regularization term (equation (16)) might help improve the performance in the flows and generative modeling applications: it would be interesting to regularize the other methods to study the impact.**
> >
> > **[Superior performance of ASWD depends on the application]** As we summarised in Response point (5), the ASWD outperforms the state-of-the-art slice-based Wasserstein distance metrics, including GSWD and DSWD, in 15 out of 17 experiment setups in gradient flow, generative models, and sliced Wasserstein auto-encoder examples with quantitative results, in the same examples where these algorithms reported their state-of-the-art performance. In experiments such as sliced Wasserstein barycenter and color transferring where qualitative results are presented, the ASWD produces visually smooth and realistic images and meaningful barycenters. These were achieved with a simple injective network architecture throughout all experiments with minimal parameter tuning. In contrast, the best performance from GSWD or DSWD/DGSWD (with any choice of defining functions) is observed in 2 out of 17 experiment setups while requiring hand-picking the best-performing defining function which is essentially a hyperparameter in the GSWD or DSWD/DGSWD that is hard to optimize.
> >
> > **[Regularization]** We want to clarify that the regularization term (Equation (16)) does not apply to other slice-based Wasserstein metrics. Specifically, the regularization term is applied to constrain the parameterized mapping $g_\omega(\cdot)$, while this mapping is not used in other methods. [2] proposed the DSWD to regularize the distribution of unit vectors $\theta$, and we have included the DSWD in our experiment section.
> >
> > (9)  **Hence, I encourage the authors to provide more significant evidence on why ASWD is a better alternative to SWD than GSWD, or to show a more nuanced picture in light of the empirical results in the appendix; for now, I am not convinced by the superiority of ASWD over SWD/GSWD given the motivation (in particular, l.234-241) nor the theoretical and empirical contributions.**
> >
> > We hope that our responses in Response points (5) and (8) above would address your concern. Theoretically, the ASWD is a valid distance metric with data-adatpve nonlinear projections that can be end-to-end optimized; while the GSWD has a limited choice of defining functions that cannot be end-to-end optimized and the GSWD-NN is not a valid distance metric. Empirical performance-wise, with a simple injective network structure adopted for ASWD throughout all experiments, we show that the ASWD achieves the best performance in 15 out of 17 experiment setups where quantitative results are presented, in diverse problem domains including the experiment setup where the GSWD reported its superior performance. In comparison, any particular form of GSWD or DSWD (with a particular defining function) achieves the best performance in 1 out of 17 quantitative sub-experiments, and even the best performance from the GSWD or DSWD (with any choice of defining functions) achieves the best performance in 2 out of the 17 experiment setups. And in experiments with qualitative results such as color transfer and barycenter generation, the ASWD produces visually more smooth and realistic images and meaningful barycenters compared with the GSWD. We appreciate that GSWD is a seminal work with impressive performance that inspired our work, and we believe that the theoretical and empirical contributions presented here and in the paper provide significant evidence on why ASWD is a better alternative to SWD than GSWD.
> >
> > (10) **Typos and minor suggestions.**
> >
> > Thanks for your suggestions, we will revise accordingly.
> >
> > [1] S. Kolouri, K. Nadjahi, U. Simsekli, R. Badeau, and G. Rohde. Generalized sliced Wasserstein distances. In Proc. Advances in Neural Information Processing Systems (NeurIPS), pp. 261–272, Vancouver, Canada, 2019.
> >
> > [2] K. Nguyen, N. Ho, T. Pham, and H. Bui. Distributional sliced-Wasserstein and applications to generative modeling. In Proc. International Conference on Learning Representations (ICLR), Vienna, Austria, 2021.

---

> > > ### Comment · Reviewer_boZk · 2021-08-31
> > > **Updated opinion**
> > >
> > > Thank you very much for your detailed response. I am now more convinced by the superior performance of ASWD in practice, and I would like to acknowledge the effort made by the authors in deploying their method in various applications.
> > >
> > > However, the claim that ASWD is conceptually more powerful than GSWD is still unclear to me, especially given the comments of reviewer HUCr: if the implemented ASWD does not verify the requirements of Corollary 1.1., then it is even more similar to GSWD based on neural networks, which is also data-adaptive and has not yet been to yield a valid metric either.
> > >
> > > Because of the theoretical inaccuracies, I think this paper would benefit from another careful round of review, hence I am keeping my original score. That being said, the empirical study is quite dense and interesting, so I wouldn't mind if this paper gets accepted.

---

> > > > ### Author Response · Authors · 2021-09-01
> > > > **Thank you for your comment**
> > > >
> > > > Thank you so much for your valuable feedback and acknowledgement of our empirical study in this work.
> > > >
> > > > We have revised Corollary 1.1 based on the suggestion from Reviewer HUCr (see "Update on Corollary 1.1" in the response to Reviewer HUCr for details).
> > > >
> > > > We took some time adding a new proof of this revised version of Corollary 1.1, following exactly what Reviewer HUCr advised. Therefore, the numerical implementation of the ASWD is indeed guaranteed to be a valid metric, so the concern of theoretical inaccuracy is addressed.
> > > >
> > > > With this update, we sincerely hope that we have now addressed all of your concerns and you could consider adjusting your assessment.
> > > >
> > > > If there is anything else that remains unclear, we'd be more than happy to clear any doubts and concerns you might have.

---

### Official Review · Reviewer_HUCr · 2021-07-15

**Rating:** 7
**Confidence:** 4

**Summary:**

This paper introduces _Augmented Sliced Wasserstein Distances_ (ASWD) which consist, in a nutshell, in computing the usual SWD between pushed measures: $\mathrm{ASWD}(\mu,\nu; g) = \mathrm{SWD}(g \circ \mu, g \circ \nu)$ (note: the pushforward operator does not render properly, so I will use $\circ$ instead). The idea is then to maximize on $g$ (under constraint/regularization) to obtain a pushforward map $g$ that provides the most discriminative projection directions. Authors provide an exhaustive condition on $g$ (namely, injectivity) to make the ASWD a metric between probability measures. Their approach is then showcased on a various set of benchmark experiments in computational optimal transport (gradient flows, generative models, auto-encoder, barycenters, color transfert...).

**Limitations And Societal Impact:**

The paper mentions some of its limitations and potential negative social impact (which are quite general to the ML field); I did not identify specific impact this paper could have.

**Main Review:**

[Edit: increased grade 6-->7 after rebuttal]


Quality:
The paper is overall interesting and supported by a large number of convincing experiments. Some of its theoretical aspects can be improved though:
- When defining the ASWD, the map $g$ should be chosen to ensure that $g \circ \mu, \nu$ still have finite $k$-th moments.
- Corollary 1.1, in its current state, seems wrong/incomplete. It is not clear why the argmax should be reached. If my understanding is correct, it is actually likely that the supremum (even when restricted on bounded injective maps) is infinite (each instance is finite, as stated in line 496, but the supremum may not). For instance, let $\mu = \delta_x, \nu = \delta_y$ with $x,y \in \mathbb{R}^d$; then $\mathrm{ASWD}(\mu,\nu; g) = \int_\theta |\braket{\theta, g(x) - g(y)}| \mathrm{d} \theta$. Taking $g = g_\omega \simeq \omega \arctan( \cdot )$ may lead to arbitrarily large values (while such $g_\omega$ is injective and bounded, at fixed $\omega$). This is actually the point of line 199 (regularization in the NN); in my opinion, this should be the optimization problem used to define the ASWD metric (to be compared with the Vaniila-ASWD if one does not optimize on $g$), and it would be nice to have a proof that this supremum is reached, perhaps a statement about the (non-)convexity(?) of the problem, etc., as it is actually at the core of the work.

Clarity:
The paper is well-written and easy to read overall.

Originality/Significance:
This paper heavily builds on previous works (in particular those of Kolouri et al.) which it properly credits. It provides simple yet seemingly efficient variations that improve on previous results. A quick look at the code and at the variety of experiments run also suggest that, once made publicly available, these tools may be useful of the computational OT community.


Minor remarks and suggestions:
- The paper may benefits from being written in a measure-based formalism; the current definitions suggest that $\mu,\nu$ should have densities $p_\mu, p_\nu$ and thus that discrete measures are not encompassed, which is obviously not the case.
- In Figure 14, it would be nice to include (Regularized) OT barycenters (available in PythonOptimalTransport for instance) to showcase the (potential) difference between sliced approaches and the "classical one".
- (line 104) It may be worth briefly mentioning these "certain conditions" for the sake of completeness.

**Time Spent Reviewing:**

3

---

> ### Author Response · Authors · 2021-08-10
> **Response to Reviewer HUCr**
>
> Many thanks for your positive and constructive feedback! Please see below our detailed response to your comments:
>
> (1) **When defining the ASWD, the map $g$ should be chosen to ensure that $g\circ \mu, \nu$ still have finite $k$-th moments.**
>
> We agree that the mapping $g$ should be chosen to ensure that $g\circ \mu, \nu$  have finite $k$-th moments and we will add it in defining the ASWD after rebuttal. We would like to emphasise that this is a mild condition that can be easily satisfied in practice, e.g. using the regularization term defined in Equation (16) to prevent the output of $g(\cdot)$ from being arbitrarily large for empirical measures.
>
> (2) **Corollary 1.1, in its current state, seems wrong/incomplete ... as it is actually at the core of the work.**
>
> **[It is actually likely that the supremum is infinite]** We will clarify Corollary 1.1 in the revision. We omitted the parameter $\omega$ in the current form of Corollary 1.1 as the parameterization of $g(\cdot)$ was introduced later in Sec 3.2 (numerical implementation). In fact, we aim to show in Corollary 1.1 that for a mapping $g_\omega(\cdot)$ parameterized by $\omega\in \mathcal{W}$, the ASWD is a metric when $\exists M \in \mathbb{R}$ the inequality $||g_\omega(x)||_2\leq M$ holds for all $\omega\in \mathcal{W}$ and for all $x\in \mathbb{R}^d$.
> This condition can be met through the optimization of $\omega$ with the regularization term defined in Equation (16) as you rightly pointed out. As a result, the supremum is finite in this context. We will improve the structure of our paper to remove this confusion.
>
> **[Compare ASWD with the Vanilla-ASWD if one does not optimize on $g$]** Indeed, we have carried out this comparison and reported results in Appendix G.2. We observe in Fig. 5 that compared with the ASWD, the performance of the Vanilla-ASWD is much worse, indicating that the necessity of optimizing $g$.
>
> **[(Non-)convexity of the problem]** The problem is non-convex as it involves the optimization of a neural network. In practice, the mapping is optimized with a fixed number of steps for computational reasons as presented in the pseudocode provided in Appendix E. Even if the supremum is not achieved, we show empirically, e.g. in the ablation study in Appendix G.2, that the optimized hypersurfaces are informative enough to distinguish samples from different compared distributions and led to state-of-the-art results.
>
> (3) **The paper may benefits from being written in a measure-based formalism; the current definitions suggest that should have densities and thus that discrete measures are not encompassed, which is obviously not the case.**
>
> We agree that a measure-based formalism may benefit the paper and will update it accordingly. The motivation of adopting the current notation involving probability density function is to follow the practice on some of the state-of-the-art slice-based Wasserstein metric literature, such as [1] [2] [3].
>
> (4) **In Figure 14, it would be nice to include (Regularized) OT barycenters to showcase the difference between sliced approaches and the "classical one".**
>
> We will include the result with regularized OT barycenter after the rebuttal.
>
> (5) **(line 104) It may be worth briefly mentioning these "certain condition" for the sake of completeness.**
>
> We will list the conditions of a valid defining function $\beta$ rendering a bijective GRT in the revision.
>
> [1] S. Kolouri, K. Nadjahi, U. Simsekli, R. Badeau, and G. Rohde. Generalized sliced Wasserstein distances. In Proc. Advances in Neural Information Processing Systems (NeurIPS), pp. 261–272, Vancouver, Canada, 2019.
>
> [2] S. Kolouri, P. E Pope, C. E. Martin, and G. K. Rohde. Sliced-Wasserstein autoencoders. In Proc. International Conference on Learning Representations (ICLR), New Orleans, Louisiana, USA, 2019b.
>
> [3] K. Nguyen, N. Ho, T. Pham, and H. Bui. Distributional sliced-Wasserstein and applications to generative modeling. In Proc. International Conference on Learning Representations (ICLR), Vienna, Austria, 2021.

---

> > ### Comment · Reviewer_HUCr · 2021-08-23
> > **About corollary 1.1**
> >
> > Thank you for your answer.
> >
> > Regarding our discussion around Corollary 1.1 ; I agree that a version assuming that $g$ belongs to some class $\mathcal{G}$ whose elements are bounded by a *same* constant $M$ would be correct (which is a much stronger assumption). However, if my understanding is correct, this is not what happens afterward (Eq (15) and (16)) : the proposed $g_\omega : x \mapsto [x, \phi_\omega(x)]$, where $\phi_\omega$ is a neural network, does not satisfy these assumptions.
> > Thus, defining the ASWD as in Corollary 1.1 would introduce an unfortunate gap between theory (maps $g$ uniformly bounded) and practice ($g$ unbounded, parametrized by a NN that may be unbounded as well) ; I think it would be better to define the $g^*$
> >  in Corollary 1.1 through (a variant of) the optimization problem in (16) (removing the dependence on $\omega$, but clearly mentioning the class $\mathcal{G}$ to which $g$ must belong) --- while keeping the definition $\mathrm{ASWD}(\mu,\nu) = \mathrm{ASWD}(\mu,\nu,g^*)$. If I am correct, this would easily ensure that the chosen $g^*$ is well defined (and injective) without introducing unnecessary restrictive assumptions that would not be on par with the experiments.

---

> > > ### Author Response · Authors · 2021-09-01
> > > **Update on Corollary 1.1**
> > >
> > > Thank you very much for your insightful comments and suggestion. This discussion has been really helpful.
> > >
> > > We have revised Corollary 1.1 as you suggested, defining $g^*(\cdot)$ through a variant of the optimization problem in Equation (16). Besides, we have also proved that the ASWD defined with the optimized $g^*(\cdot)$ is a valid metric for $\lambda\in (1, +\infty)$. In a nutshell, we prove that the ASWD with order $k$ is upper bounded by certain types of regularisation functions. Hence, the optimization objective is non-positive and the supremum is finite so argmax should be reached. As you pointed out, this would easily ensure that the chosen $g^*(\cdot)$ is well defined without unnecessary restrictive assumptions on the class $\mathcal{G}$ to which $g$ must belong to.
> > >
> > > To make the proof simple and intuitive we derive a loose lower bound for $\lambda > 1$. A tight lower bound will depend on the dimensionality of $g(\cdot)$ and will be close to 0, which is also supported in our empirical studies.
> > >
> > > The revised Corollary 1.1 and its proof are as follows:
> > >
> > > **Corollary 1.1**
> > >
> > > The augmented sliced Wasserstein distance (ASWD) of order $k\in[1,+\infty)$ between two probability $\mu, \nu\in P_k(\mathbb{R}^d)$ defined by Equation (13) with the optimal mapping
> > >
> > > $g^*(\cdot)=\underset{g}{\arg\max}$ {$ASWD_k(\mu, \nu; g)- L(\mu,\nu; \lambda)$} **(1)**
> > >
> > > is a metric on  $P_k(\mathbb{R}^d)$, where $L(\mu,\nu; \lambda)=\lambda(\underset{x\sim \mu}{\mathbb{E}}\big[||g(x)||_2^k\big]^{\frac{1}{k}}+\underset{y\sim \nu}{\mathbb{E}}\big[||g(y)||_2^k\big]^{\frac{1}{k}})$  for $\lambda\in (1, +\infty)$.
> > >
> > > ***Proof of Corollary 1.1***
> > >
> > > We prove here that the optimal mappings obtained by solving the optimization problem $g^*(\cdot)=\underset{g}{\arg\max}${$ASWD_k(\mu, \nu; g)- L(\mu,\nu; \lambda)$} satisfies $||g^*(x)||_2 < \infty$ for $\forall x \in \mathbb{R}^d \sim \mu$ and $\forall x  \in \mathbb{R}^d \sim \nu$.
> > >
> > > Recall that in Equation (14) in the paper, the ASWD can be rewritten as:
> > >
> > > $ASWD_k(\mu, \nu; g)=\text{SWD}_k(\mu_g, \nu_g)$
> > >
> > > $=\bigg( \int_{\mathbb{S}^{d_\theta-1}} W_k^k (\mathcal{R}p_{\mu_g}(\cdot, \theta), \mathcal{R}p_{\nu_g}(\cdot, \theta))d\theta \bigg)^{\frac{1}{k}}$, **(2)**
> > >
> > > where transformed variables $\hat{x}=g(x) \sim \mu_g$ for $x\sim\mu$, $\hat{y}=g(y) \sim \nu_g$ for $y \sim \nu$, respectively.
> > > Combining  the equation above with Equation (2) in the paper:
> > >
> > >
> > > $ASWD_k(\mu, \nu; g)
> > >     	=\bigg( \int_{\mathbb{S}^{d_\theta-1}} W_k^k (\mathcal{R}p_{\mu_g}(\cdot, \theta), \mathcal{R}p_{\nu_g}(\cdot, \theta))d\theta \bigg)^{\frac{1}{k}} =\bigg( \int_{\mathbb{S}^{d_\theta-1}} \int_0^1 |F_{P_{\theta}\\#p_{\mu_g}}^{-1}(z)-F_{P_{\theta}\\#p_{\nu_g}}^{-1}(z)|^k dz d\theta \bigg)^{\frac{1}{k}}$
> > >
> > > $\leq \bigg(\int_{\mathbb{S}^{d_\theta-1}} \int_0^1 \big(|F_{R_\theta(\mu_g)}^{-1}(z)|+|F_{R_\theta(\nu_g)}^{-1}(z)|\big)^k dz d\theta \bigg)^{\frac{1}{k}}$,  **(3)**
> > >
> > > where  $\\#$ denotes the push forward operator, $P_{\theta}: x\in\mathbb{R}^{d_\theta}\rightarrow\langle x, \theta \rangle\in \mathbb{R}$, and $R_\theta(\mu_g)=P_{\theta}\\#p_{\mu_g}$, $R_\theta(\nu_g)=P_{\theta}\\#p_{\nu_g}$  refer to one-dimensional measures obtained by slicing $\mu_g, \nu_g$ with a unit vector $\theta$, $F_{R_\theta(\mu_g)}^{-1}$ and $F_{R_\theta(\nu_g)}^{-1}$ are inverse cumulative distribution functions (CDFs) of $R_\theta(\mu_g)$ and $R_\theta(\nu_g)$, respectively.
> > >
> > > By repeatedly applying the Minkowski's inequality to Equation (3), we obtain the following inequalities:
> > >
> > > $ASWD_k(\mu, \nu; g)\leq\bigg(\int_{\mathbb{S}^{d_\theta-1}} \int_0^1 \big(|F_{R_\theta(\mu_g)}^{-1}(z)|+|F_{R_\theta(\nu_g)}^{-1}(z)|\big)^k dz d\theta \bigg)^{\frac{1}{k}}$
> > >
> > > $\leq\bigg( \int_{\mathbb{S}^{d_\theta-1}} \bigg[\bigg(\int_0^1 |F_{R_\theta(\mu_g)}^{-1}(z)|^k dz \bigg)^{\frac{1}{k}}+\bigg(\int_0^1|F_{R_\theta(\nu_g)}^{-1}(z)|^k dz\bigg)^{\frac{1}{k}} \bigg]^k d\theta \bigg)^{\frac{1}{k}}$.
> > >
> > > $\leq\bigg[\int_{\mathbb{S}^{d_\theta-1}}\int_0^1 |F_{R_\theta(\mu_g)}^{-1}(z)|^k dz d{\theta}\bigg]^{\frac{1}{k}}+\bigg[\int_{\mathbb{S}^{d_\theta-1}}\int_0^1 |F_{R_\theta(\nu_g)}^{-1}(z)|^k dz d{\theta}\bigg]^{\frac{1}{k}}$. **(4)**
> > >
> > > Let $s=\langle \hat{x}, \theta\rangle$, then $z=F_{R_\theta(\mu_g)}(s)$, $dz=d F_{R_\theta(\mu_g)}(s)$:
> > >
> > > $\int_0^1 |F_{R_\theta(\mu_g)}^{-1}(z)|^kdz$
> > > $=\int_\mathbb{R} |s|^k dF_{R_\theta(\mu_g)}(s)$
> > >
> > > $=\int_{\mathbb{R}^{d_\theta}} |\langle \hat{x},\theta \rangle|^k d\mu_g$
> > >
> > > $=\int_{\mathbb{R}^d} |\langle g(x),\theta \rangle|^k d{\mu},$ **(5)**
> > >
> > > where the last two equations are obtained through the definitions of the push-forward operators. Therefore, the following inequalities hold:
> > >
> > > $ASWD_k(\mu, \nu; g)\leq\bigg[\int_{\mathbb{S}^{d_\theta-1}}\int_0^1 |F_{R_\theta(\mu_g)}^{-1}(z)|^k dz d{\theta}\bigg]^{\frac{1}{k}}$
> > > $+\bigg[\int_{\mathbb{S}^{d_\theta-1}}\int_0^1 |F_{R_\theta(\nu_g)}^{-1}(z)|^k dzd{\theta}\bigg]^{\frac{1}{k}}$
> > >
> > > $=\bigg[\int_{\mathbb{S}^{d_\theta-1}}\int_{\mathbb{R}^d} |\langle g(x),\theta \rangle|^k d\mu d{\theta}\bigg]^{\frac{1}{k}}+\bigg[\int_{\mathbb{S}^{d_\theta-1}}\int_{\mathbb{R}^d} |\langle g(y),\theta \rangle|^k d\nu d{\theta}\bigg]^{\frac{1}{k}}\leq\underset{x\sim \mu}{\mathbb{E}}\big[||g(x)||^k_2 \big]^{\frac{1}{k}}+\underset{y\sim\nu}{\mathbb{E}}\big[||g(y)||^k_2 \big]^{\frac{1}{k}}$ **(6)**
> > >
> > > Then we obtain the following inequalities for the optimization objective:
> > >
> > > $ASWD_k(\mu, \nu; g)- L(\mu,\nu; \lambda)$ $\leq\big(\underset{x\sim\mu}{\mathbb{E}}\big[||g(x)||^k_2 \big]^{\frac{1}{k}}+\underset{y\sim\nu}{\mathbb{E}}\big[||g(y)||^k_2\big]^{\frac{1}{k}}\big)-\lambda(\underset{x\sim\mu}{\mathbb{E}}\big[||g(x)||^k_2\big]^{\frac{1}{k}}+\underset{y\sim\nu}{\mathbb{E}}\big[||g(y)||^k_2\big]^{\frac{1}{k}})$
> > >
> > > $=\big(1-\lambda\big)\big(\underset{x\sim\mu}{\mathbb{E}}\big[||g(x)||^k_2 \big]^{\frac{1}{k}}+\underset{y\sim\nu}{\mathbb{E}}\big[||g(y)||^k_2\big]^{\frac{1}{k}}\big)$ **(7)**
> > >
> > > When we set $\lambda\in(1, +\infty)$, if $\exists x\in\mathbb{R}^d \sim \mu$ or $y\in\mathbb{R}^d \sim \nu$ such that $||g(x)||_2\rightarrow\infty$ or $||g(y)||_2\rightarrow\infty$, the optimization objective approaches negative infinity, implying $g(\cdot)$ is not the optimal mapping. Therefore, by adopting Equation (1) as the optimization objective, the optimal $g^*(\cdot)$ is confined to the set of functions $\mathcal{G}=\{g(x): \mathbb{R}^d\rightarrow\mathbb{R}^{d_\theta}\}$ satisfying $||g^*(x)||_2<\infty$ for $\forall x \in \mathbb{R}^d \sim \mu$ and $\forall x \in \mathbb{R}^d \sim \nu$.
> > >
> > > $\text{ASWD}_k(\mu, \nu; g^*)$ is hence finite due to Equation (2). Given that we have proved that the ASWD is finite with the optimal $g^*(\cdot)$, the proof of the symmetry, triangle inequality, identify of indiscernibles, and non-negativity follows the original proof provided in Appendix D.
> > >
> > > ***END OF PROOF***
> > >
> > > **Remark:** It is worth noting that $\lambda > 1$ in Corollary 1.1 is a sufficient condition for the supremum of the optimization objective to be non-positive and $||g^*(x)||_2 < \infty$ for $\forall x \in \mathbb{R}^d \sim \mu$ and $\forall x \in \mathbb{R}^d \sim \nu$. In practice, $0 < \lambda\leq 1$ may also lead to valid metrics. Specifically, the upper bound of the optimization objective given in Equation (6) is obtained by applying:
> > >
> > > $|\langle g(x),\theta \rangle|=||g(x)||_2|\cos(\alpha)|\leq||g(x)||_2$, **(8)**
> > >
> > > where $\alpha$ is the angle between $\theta$ and $g(x)$. In high-dimensional spaces, Equation (8) gives a very loose bound since in high-dimensional spaces the majority of sampled $\theta$ would be nearly orthogonal to $g(x)$ and $\cos(\alpha)$ is nearly zero with high probability [1]. Empirically we found that across all the experiment results, $\lambda$ in a candidate set of {0.01, 0.05, 0.1, 0.5, 1, 10, 100} all lead to finite $g^*(\cdot)$.
> > >
> > > [1] S. Kolouri, K. Nadjahi, U. Simsekli, R. Badeau, and G. Rohde. Generalized sliced Wasserstein distances. In Proc. Advances in Neural Information Processing Systems (NeurIPS), pp. 261–272, Vancouver, Canada, 2019.

---

> > > > ### Comment · Reviewer_HUCr · 2021-09-01
> > > > **Nice**
> > > >
> > > > Thank you for this detailed and instructive comment. I recognize your effort to improve Corollary 1.1, and this version seems much more relevant to me.
> > > >
> > > > I quickly checked the intermediate calculations, which seem correct. Regarding the mathematical reasoning, by $\|g(x)\| \to \infty$ (where $x$ is fixed), do you mean that $\sup_{g \in \mathcal{G}} \|g(x)\| = \infty$ ?
> > > >
> > > > I think the claim/proof could be slightly more clear if you said something like: Let $g = 0$, then $ASWD(\mu,\nu,0) - \underbrace{L(\mu,\nu,\lambda; g=0)}_{=0} =: M$ which is a lower bound of the maximal value your objective function can reach. And from this (and the calculation you have made), deduce that you can restrict to maps $g$ in $\mathcal{G} = \{ g,\ \forall x, \|g(x)\| \lesssim M \}$ ; point being that $M$ would be uniform over all $g$ there. I think this is more or less there in your current proof, but it is not very striking to me.
> > > >
> > > > Note: it may also be helpful to make explicit the dependence of $L$ on $g$ (e.g., $L(\mu,\nu,\lambda; g)$.

---

> > > > > ### Author Response · Authors · 2021-09-01
> > > > > **Clarification**
> > > > >
> > > > > Thank you very much for your prompt feedback.
> > > > >
> > > > > For $||g(x)||_2\rightarrow \infty$, we intend to use proof by
> > > > > contradiction to show that the optimal $||g^*(x)||_2$ is finite for $\forall x\in\mathbb{R}^d \sim \mu$. To avoid the confusion, we will change the part of proof to Corollary 1.1 in our last response as follows:
> > > > >
> > > > > from
> > > > >
> > > > >  "When we set $\lambda\in(1, +\infty)$, if $\exists x\in\mathbb{R}^d \sim \mu$ or $y\in\mathbb{R}^d \sim \nu$ such that $||g(x)||_2\rightarrow\infty$ or $||g(y)||_2\rightarrow\infty$, the optimization objective approaches negative infinity, implying $g(\cdot)$ is not the optimal mapping.''
> > > > >
> > > > >  to
> > > > >
> > > > >  "When we set $\lambda\in(1, +\infty)$, if $\exists x\in\mathbb{R}^d \sim \mu$ or $y\in\mathbb{R}^d \sim \nu$ such that $||g^*(x)||_2\rightarrow\infty$ or $||g^*(y)||_2\rightarrow\infty$, the optimization objective approaches negative infinity, implying $g^*(\cdot)$ is not the optimal mapping.''
> > > > >
> > > > > We thank your suggestion for improving the clarity of proof with a specific lower bound of the optimization objective. However, the ASWD requires $g(\cdot)$ to be injective, so $g$ cannot be made $0$ to meet this requirement.
> > > > >
> > > > > It will indeed be helpful to make the dependence of $L$ on $g$ explicit. We thank your suggestion and will adopt the notation $L(\mu, \nu \lambda; g)$.
> > > > >
> > > > > As always, we are more than happy to clarify further if you have more questions.

---

> > > > > > ### Comment · Reviewer_HUCr · 2021-09-02
> > > > > > **Indeed**
> > > > > >
> > > > > > Indeed, $g=0$ was a poor suggestion, sorry about that. I was just thinking of "some explicit map that would satisfy $\mathbb{E}_{X \sim \mu}[\|g(X)\|^2] < \infty$ (resp.~$\nu$), using it as a lower bound. But I think your proof by contradiction is mostly correct as well.
> > > > > >
> > > > > > Note: you need to make sure that $g^*$ is still injective, which may not be obvious as this is not a "closed" property, i.e., all element in $\mathcal{G}$ may be injective, but $\mathrm{argmax} \{g \in \mathcal{G}, ...\}$ may not. I may be missing something though.
> > > > > >
> > > > > > Given the considerable efforts made to clarify this proof along with detailed reviews provided to other reviewers, I will improve my grade.
> > > > > >
> > > > > > I encourage the authors to thoroughly revise their work taking the various reviewers remarks into account (should it be the camera-ready version if acceptance, or a re-submission otherwise) and, if possible, to ask an external reader to proofread the mathematical content (I am confident that proofs should be substantially correct, but given the large amount of modifications required since the submitted version, I am worried that some inaccuracies may remain/be introduced).

---

> > > > > > > ### Author Response · Authors · 2021-09-02
> > > > > > > **Thank you**
> > > > > > >
> > > > > > > Thank you so much for your feedback and raising your score. We are grateful to have this fruitful discussion to make the paper scientifically convincing and technically sound supported by the extensive empirical study. We are thoroughly revising the work taking into account all reviewers’ comments for the camera-ready version if the paper is accepted. We will also ask an external field expert to proofread the mathematical contents. Once again, we very much appreciate your time and valuable suggestions.

---

### Author Response · Authors · 2021-08-10
**Response to all reviewers**

We would like to thank all reviewers for their insightful suggestions to improve this paper and their appreciation of the paper and the proposed method, and we will reflect many valuable suggestions made by the reviewers after the rebuttal. We would like to reiterate our key contributions before responding to each reviewer individually.

At a high level, this work proposes a new family of slice-based Wasserstein distance metrics with a novel incorporation of injective neural networks to learn nonlinear projections that can capture the complex structure of the data distributions. We address a major limitation of the state-of-the-art approaches in incorporating nonlinear projections into slice-based Wasserstein distances, i.e. the choice of defining functions are limited, problem specific, and is essentially a hyper-parameter that cannot be end-to-end optimized for constructing a valid distance metric. Compared with previous work, the ASWD is data-adaptive, can be end-to-end optimised, is a valid distance metric, and has been shown to lead to strong empirical performance in a wide range of benchmark experiments. We are thankful that our work has been appreciated by all reviewers as "interesting and supported by a large number of convincing experiments" (Reviewer HUCr), "can be easily applied to various classical machine learning applications" (Reviewer boZk), "easy to implement and understand"( Reviewer 7Tc3) and "convincing" (Reviewer xvVx).

---

### Decision · Program_Chairs · 2021-09-27

**Decision:**

Reject

**Comment:**

This paper received four reviews, with scores scattered around the acceptance threshold. All the reviewers basically evaluated the main idea of this paper interesting, and clarity of the presentation of the main idea positively, whereas evaluations on significance were somehow mixed. Most notably, the reviewers raised a significant number of criticisms on several aspects of the contents of this paper, and the authors seem to have responded to most of them with prospects of how they will revise the paper. Some comments were then followed by further discussion between the authors and the reviewers. All of these should be quite fruitful in further improving the paper, as evidenced by some reviewers raising their scores. At the same time, it suggests that the submitted version which we reviewed had a large room for significant improvements, and some reviewers, in the discussion among us, did show their concerns as to whether the planned revision will eventually yield a paper with sufficient quality. I therefore think that another round of reviewing should be necessary to judge the technical quality of this contribution after the planned extensive revision, so that I would not be able to recommend acceptance of this paper at this stage.

An additional minor point: Equation (39) in Appendix D seems incorrect, but equation (40) should be correct, as it has been proved in page 15, under the heading "Triangle inequality". Simply removing the right-hand side of equation (39) should fix this defect.